# ROBUSTNESS REPROGRAMMING FOR REPRESENTATION LEARNING

**Zhichao Hou[1], MohamadAli Torkamani[2], Hamid Krim[1], Xiaorui Liu[1]**
[1]North Carolina State University  [2]Amazon Web Services
`zhou4@ncsu.edu, alitor@amazon.com, ahk@ncsu.edu, xliu96@ncsu.edu`

## ABSTRACT

This work tackles an intriguing and fundamental open challenge in representation learning: Given a well-trained deep learning model, can it be reprogrammed to enhance its robustness against adversarial or noisy input perturbations without altering its parameters? To explore this, we revisit the core feature transformation mechanism in representation learning and propose a novel non-linear robust pattern matching technique as a robust alternative. Furthermore, we introduce three model reprogramming paradigms to offer flexible control of robustness under different efficiency requirements. Comprehensive experiments and ablation studies across diverse learning models ranging from basic linear model and MLPs to shallow and modern deep ConvNets demonstrate the effectiveness of our approaches. This work not only opens a promising and orthogonal direction for improving adversarial defenses in deep learning beyond existing methods but also provides new insights into designing more resilient AI systems with robust statistics. Our implementation is available at https://github.com/chris-hzc/Robustness-Reprogramming.

## 1 INTRODUCTION

Deep neural networks (DNNs) have made significant impacts across various domains due to their powerful capability of learning representation from high-dimensional data (LeCun et al., 2015; Goodfellow et al., 2016). However, it has been well-documented that DNNs are highly vulnerable to adversarial attacks (Szegedy, 2013; Biggio et al., 2013). These vulnerabilities are prevalent across various model architectures, attack capacities, attack knowledge, data modalities, and prediction tasks, which hinders their reliable deployment in real-world applications due to potential economic, ethical, and societal risks (AI, 2023).

In light of the burgeoning development of AI, the robustness and reliability of deep learning models have become increasingly crucial and of particular interest. A mount of endeavors attempting to safeguard DNNs have demonstrated promising robustness, including robust training (Madry, 2017; Zhang et al., 2019; Gowal et al., 2021; Li & Liu, 2023), regularization (Cisse et al., 2017; Zheng et al., 2016), and purification techniques (Ho & Vasconcelos, 2022; Nie et al., 2022; Shi et al., 2021; Yoon et al., 2021). However, these methods often suffer from catastrophic pitfalls like cumbersome training processes or domain-specific heuristics, failing to deliver the desired robustness gains in an efficient and adaptable manner.

Despite numerous advancements in adversarial defenses, an open challenge persists: *Is it possible to reprogram a well-trained model to achieve the desired robustness without modifying its parameters?* This question is of particular significance in the current era of large-scale models. Reprogramming without training is highly efficient, as the pretraining-and-finetuning paradigm grants access to substantial open source pre-trained parameters, eliminating the need for additional training. Moreover, reprogramming offers an innovative, complementary, and orthogonal approach to existing defenses, paving the way to reshape the landscape of robust deep learning.

To address this research gap, we firstly delve deeper into the basic neurons of DNNs to investigate the fundamental mechanism of representation learning. At its core, the linear feature transformations serve as an essential building block to capture particular feature patterns of interest in most deep learning models. For instance, the Multi-Layer Perceptron (MLP) fundamentally consists of

multiple stacked linear mapping layers and activation functions; the convolution operations in Convolution Neural Networks (CNNs) (He et al., 2016) execute a linear feature mapping over local patches using the convolution kernels; the attention mechanism in Transformers (Vaswani, 2017) performs linear transformations over the contextualized token vectors. This linear feature mapping functions as *Linear Pattern Matching* by capturing the certain patterns that are highly correlated with the model parameters. However, this pattern matching manner is highly sensitive to data perturbations, which explains the breakdown of the deep learning models under the adversarial environments.

To this end, we propose a novel approach, *Nonlinear Robust Pattern Matching*, which significantly improves the robustness while maintaining the feature pattern matching behaviors. Furthermore, we also introduce a flexible and efficient strategy, *Robustness Reprogramming*, which can be deployed under three paradigms to improve the robustness, accommodating varying resource constraints and robustness requirements. This innovative framework promises to redefine the landscape of robust deep learning, paving the way for enhanced resilience against adversarial threats.

Our contributions can be summarized as follows: (1) We propose a new perspective on representation learning by formulating *Linear Pattern Matching* (the fundamental mechanism of feature extraction in deep learning) as ordinary least-square problems; (2) Built upon our novel viewpoint, we introduce *Nonlinear Robust Pattern Matching* as an alternative robust operation and provide theoretical convergence and robustness guarantees for its effectiveness; (3) We develop an innovative and adaptable strategy, *Robustness Reprogramming*, which includes three progressive paradigms to enhance the resilience of given pre-trained models; and (4) We conduct comprehensive experiments to demonstrate the effectiveness of our proposed approaches across various backbone architectures, using multiple evaluation methods and providing several insightful analyses.

## 2 RELATED WORKS

**Adversarial Attacks.** Adversarial attacks can generally be categorized into two types: white-box and black-box attacks. In white-box attacks, the attacker has complete access to the target neural network, including its architecture, parameters, and gradients. Examples of such attacks include gradient-based methods like FGSM (Goodfellow et al., 2014), DeepFool (Moosavi-Dezfooli et al., 2016), PGD (Madry, 2017), and C&W attacks (Carlini & Wagner, 2017). On the other hand, black-box attacks do not have full access to the model's internal information; the attacker can only use the model's input and output responses. Examples of black-box methods include surrogate model-based method (Papernot et al., 2017), zeroth-order optimization (Chen et al., 2017), and query-efficient methods(Andriushchenko et al., 2020; Alzantot et al., 2019). Additionally, AutoAttack (Croce & Hein, 2020b), an ensemble attack that includes two modified versions of the PGD attack, a fast adaptive boundary attack (Croce & Hein, 2020a), and a black-box query-efficient square attack (Andriushchenko et al., 2020), has demonstrated strong performance and is often considered as a reliable benchmark for evaluating adversarial robustness.

**Adversarial Defenses.** Numerous efforts have been made to enhance the robustness of deep learning models, which can broadly be categorized into empirical defenses and certifiable defenses. Empirical defenses focus on increasing robustness through various strategies: robust training methods (Madry, 2017; Zhang et al., 2019; Gowal et al., 2021; Li & Liu, 2023) introduce adversarial perturbations into the training data, while regularization-based approaches (Cisse et al., 2017; Zheng et al., 2016) stabilize models by constraining the Lipschitz constant or spectral norm of the weight matrix. Additionally, detection techniques Metzen et al. (2017); Feinman et al. (2017); Grosse et al. (2017) aim to defend against attacks by identifying adversarial inputs. Purification-based approaches seek to eliminate the adversarial signals before performing downstream tasks (Ho & Vasconcelos, 2022; Nie et al., 2022; Shi et al., 2021; Yoon et al., 2021). Recently, some novel approaches have emerged by improving robustness from the perspectives of ordinary differential equations (Kang et al., 2021; Li et al., 2022; Yan et al., 2019) and generative models (Wang et al., 2023; Nie et al., 2022; Rebuffi et al., 2021). Beyond empirical defenses, certifiable defenses (Cohen et al., 2019; Gowal et al., 2018; Fazlyab et al., 2019) offer theoretical guarantees of robustness within specific regions against any attack. However, many of these methods suffer from significant overfitting issues or depend on domain-specific heuristics, which limit their effectiveness and adaptability in achieving satisfying robustness. Additionally, techniques like robust training often entail high computational and training costs, especially when dealing with diverse noisy environments, thereby limiting their

scalability and flexibility for broader applications. The contribution of robustness reprogramming in this work is fully orthogonal to existing efforts, and they can be integrated for further enhancement.

## 3 ROBUST NONLINEAR PATTERN MATCHING

In this section, we begin by exploring the vulnerability of representation learning from the perspective of pattern matching and subsequently introduce a novel robust feature matching in Section 3.1. Following this, we develop a Newton-IRLS algorithm, which is unrolled into robust layers in Section 3.2. Lastly, we present a theoretical robustness analysis of this architecture in Section 3.3.

**Notation.** Let the input features of one instance be represented as $\boldsymbol{x} = (x_1, \ldots, x_D)^\top \in \mathbb{R}^D$, and the parameter vector as $\boldsymbol{a} = (a_1, \ldots, a_D)^\top \in \mathbb{R}^D$, where $D$ denotes the feature dimension. For simplicity, we describe our method in the case where the output is one-dimensional, i.e., $z \in \mathbb{R}$.

### 3.1 A NEW PERSPECTIVE OF REPRESENTATION LEARNING

| | Linear Pattern Matching (LPM) | Nonlinear Robust Pattern Matching (NRPM) |
|---|---|---|
| **Formula** | $z_{LPM} = \sum_{d=1}^{D} a_d x_d$ | $z_{NRPM}^{(k+1)} = D \cdot \dfrac{\sum_{d=1}^{D} w_d^{(k)} a_d x_d}{\sum_{d=1}^{D} w_d^{(k)}}$ |
| **Optimization** | $\min_{z} \sum_{d=1}^{D} (z/D - a_d x_d)^2$ | $\min_{z} \sum_{d=1}^{D} \lvert z/D - a_d x_d \rvert$ |
| **Algorithm** | Closed-form Optimal Solution | Newton-IRLS Algorithm |
| **Model (One Layer)** |  |  |

Figure 1: Vanilla Linear Pattern Matching (LPM) vs. Nonlinear Robust Pattern Matching (NRPM).

Fundamentally, DNNs inherently function as representation learning modules by transforming raw data into progressively more compact embeddings (LeCun et al., 2015; Goodfellow et al., 2016). The linear feature transformation, $z = \boldsymbol{a}^\top \boldsymbol{x} = \sum_{d=1}^{D} a_d \cdot x_d$, is the essentially building block of deep learning models to capture particular feature patterns of interest. Specifically, a certain pattern $\boldsymbol{x}$ can be captured and matched once it is highly correlated with the model parameter $\boldsymbol{a}$.

Despite the impressive capability of linear operator in enhancing the representation learning of DNNs, the vanilla deep learning models have been validated highly vulnerable (Szegedy, 2013; Biggio et al., 2013). Existing approaches including robust training and regularization techniques (Madry, 2017; Cisse et al., 2017; Zheng et al., 2016) aim to improve the robustness of feature transformations by constraining the parameters $\boldsymbol{a}$ with particular properties. However, these methods inevitably alter the feature matching behaviors, often leading to clean performance degradation without necessarily achieving improved robustness.

Different from existing works, we aim to introduce a novel perspective by exploring how to design an alternative feature mapping that enhances robustness while maximally preserving feature matching behaviors. First, we formulate the linear feature pattern matching as the optimal closed-form solution of the following problem:

$$\min_{z \in \mathbb{R}} \mathcal{L}(z) = \sum_{d=1}^{D} \left( \frac{z}{D} - a_d \cdot x_d \right)^2,$$

where the first-order optimality condition $\frac{\partial \mathcal{L}(z)}{\partial z} = 0$ yields the linear transformation $z^* = \sum_{d=1}^{D} a_d \cdot x_d$. Since this estimation is highly sensitive to outlying values due to the quadratic penalty, we propose to derive a robust alternative inspired by the Least Absolute Deviation (LAD) estimation:

$$\min_{z \in \mathbb{R}} \mathcal{L}(z) = \sum_{d=1}^{D} \left| \frac{z}{D} - a_d \cdot x_d \right|. \tag{1}$$

By replacing the quadratic penalty with a linear alternative on the residual $z/D - a_d \cdot x_d$, the impact of outliers can be significantly mitigated according to robust statistics (Huber & Ronchetti, 2011).

**Methodology.** The NRPM architecture draws significant inspirations from *optimization-induced deep learning architectures* (Ma et al., 2021; Fan et al., 2022; Hou et al., b; Liu et al., 2021b;a; Hou et al., a), where robust alternatives are derived from robust optimization formulations. Specifically, it reinterprets the linear feature pattern matching in the backbone models as solution to well-defined optimization objective, and corresponding optimization algorithm is developed to solve the problem induced by the nonlinear robust pattern matching. This approach highlights a promising direction for designing principled and inherently robust deep learning architectures.

### 3.2 Algorithm Development and Analysis

Although the LAD estimator offers robustness implication, the non-smooth objective in Eq. (1) poses a challenge in designing an efficient algorithm to be integrated neural network layers. To this end, we leverage the Newton Iterative Reweighted Least Squares (Newton-IRLS) algorithm to address the non-smooth nature of the absolute value operator $| \cdot |$ by optimizing an alternative smoothed objective function $\mathcal{U}$ with Newton method. In this section, we will first introduce the localized upper bound $\mathcal{U}$ for $\mathcal{L}$ in Lemma 3.1, and then derive the Newton-IRLS algorithm to optimize $\mathcal{L}$.

**Lemma 3.1.** *Let $\mathcal{L}(z)$ be defined in Eq. (1), and for any fixed point $z_0$, $\mathcal{U}(z, z_0)$ is defined as*

$$\mathcal{U}(z, z_0) = \sum_{d=1}^{D} w_d \cdot (a_d x_d - z/D)^2 + \frac{1}{2}\mathcal{L}(z_0), \tag{2}$$

*where $w_d = \frac{1}{2|a_d x_d - z_0/D|}$. Then, for any $z$, the following holds:*

$$(1) \, \mathcal{U}(z, z_0) \geq \mathcal{L}(z), \quad (2) \, \mathcal{U}(z_0, z_0) = \mathcal{L}(z_0).$$

*Proof.* Please refer to Appendix B.1. $\square$

The statement (1) indicates that $\mathcal{U}(z, z_0)$ serves as an upper bound for $\mathcal{L}(z)$, while statement (2) demonstrates that $\mathcal{U}(z, z_0)$ equals $\mathcal{L}(z)$ at point $z_0$. With fixed $z_0$, the alternative objective $\mathcal{U}(z, z_0)$ in Eq. (2) is quadratic and can be efficiently optimized. Therefore, instead of minimizing the non-smooth $\mathcal{L}(z)$ directly, the Newton-IRLS algorithm will obtain $z^{(k+1)}$ by optimizing the quadratic upper bound $\mathcal{U}(z, z^{(k)})$ with second-order Newton method:

$$z^{(k+1)} = D \cdot \frac{\sum_{d=1}^{D} w_d^{(k)} a_d x_d}{\sum_{d=1}^{D} w_d^{(k)}} \tag{3}$$

where $w_d^{(k)} = \frac{1}{|a_d x_d - z^{(k)}/D|}$. Please refer to Appendix B.2 for detailed derivation. As a consequence of Lemma 3.1, we can conclude the iteration $\{z^{(k)}\}_{k=1}^{K}$ fulfill the loss descent of $\mathcal{L}(z)$:

$$\mathcal{L}(z^{(k+1)}) \leq \mathcal{U}(z^{(k+1)}, z^{(k)}) \leq \mathcal{U}(z^{(k)}, z^{(k)}) = \mathcal{L}(z^{(k)}).$$

This implies Eq. (3) can achieve convergence of $\mathcal{L}$ by optimizing the localized upper bound $\mathcal{U}$.

**Implementation.** The proposed non-linear feature pattern matching is expected to improve the robustness against data perturbation in any deep learning models by replacing the vanilla linear feature transformation. In this paper, we illustrate its use cases through MLPs and convolution models. We provide detailed implementation techniques in Appendix A. Moreover, we will demonstrate how to leverage this technique for robustness reprogramming for representation learning in Section 4.

## 3.3 Theoretical Robustness Analysis

In this section, we conduct a theoretical robustness analysis comparing the vanilla Linear Pattern Matching (LPM) architecture with our Nonlinear Robust Pattern Matching (NRPM) based on influence function (Law, 1986). For simplicity, we consider a single-step case for our Newton-IRLS algorithm ($K = 1$). Denote the weighted feature random variable as $X$ and corresponding empirical distribution as $F(X) = \frac{1}{D} \sum_{d=1}^{D} \mathbb{I}_{\{X = a_d x_d\}}$. Then we can represent LPM as $z_{LPM} := T_{LPM}(F) = \sum_{d=1}^{D} a_d x_d$ and NRPM as $z_{NRPM} := T_{NRPM}(F) = D \cdot \frac{\sum_{d=1}^{D} w_d a_d x_d}{\sum_{d=1}^{D} w_d}$, where $w_d = \frac{1}{|a_d x_d - z_{LPM}/D|}$. We derive their influence functions in Theorem 3.2 to demonstrate their sensitivity against input perturbations, with a proof presented in Appendix B.3.

**Theorem 3.2** (Robustness Analysis via Influence Function). *The influence function is defined as the sensitivity of the estimate to a small contamination at $\Delta x$:*

$$IF(\Delta x; T, F) = \lim_{\epsilon \to 0} \frac{T(F_\epsilon) - T(F)}{\epsilon}$$

*where the contaminated distribution becomes $F_\epsilon = (1 - \epsilon)F + \epsilon \delta_{\Delta x}$, where $\delta_{\Delta x}$ is the Dirac delta function centered at $\Delta x$ and $F$ is the distribution of $x$. Then, we have:*

$$IF(\Delta x; T_{LPM}, F) = D(\Delta x - z_{LPM}/D),$$

*and*

$$IF(\Delta x; T_{NRPM}, F) = \frac{D w_\Delta x \left( \Delta x - z_{NRPM}/D \right)}{\sum_{d=1}^{D} w_d} \text{ where } w_{\Delta x} = \frac{1}{|\Delta x - z_{LPM}/D|}.$$

Theorem 3.2 provide several insights into the robustness of LPM and NRLPM models:

- For LPM, the influence function is given by $D(\Delta x - y) = D(\Delta x - z_{LPM}/D)$, indicating that the sensitivity of LPM depends on the difference between the perturbation $\Delta x$ and the average clean estimation $z_{LPM}/D$.

- For NRPM, the influence function is $\frac{D w_{\Delta x}(\Delta x - z_{NRPM}/D)}{\sum_{d=1}^{D} w_d}$, where $w_{\Delta x} = \frac{1}{|\Delta x - z_{LPM}/D|}$. Although the robustness of NRPM is affected by the difference $\Delta x - z_{NRPM}/D$, the influence can be significantly mitigated by the weight $w_{\Delta x}$, particularly when $\Delta x$ deviates from the average estimation $z_{LPM}/D$ of the clean data.

These analyses provide insight and explanation for the robustness nature of the proposed technique.

## 4 Robustness Reprogramming

In this section, it is ready to introduce the robustness programming techniques based on the nonlinear robust pattern matching (NRPM) derived in Section 3.1. One naive approach is to simply replace the vanilla linear pattern matching (LPM) with NRPM. However, this naive approach does not work well in practice, and we propose three robustness reprogramming paradigms to improve the robustness, accommodating varying resource constraints and robustness requirements.

While NRPM enhances model robustness, it can lead to a decrease in clean accuracy, particularly in deeper models. This reduction in performance may be due to the increasing estimation error across layers. To balance the clean-robustness performance trade-off between LPM and NRPM, it is necessary to develop a hybrid architecture as shown in Algorithm 1, where their balance is controlled by hyperparameters $\{\lambda_l\}_{l=1}^{L}$ and $L$ is the number of layers in the entire model. Based on this hybrid architecture, we propose three robustness reprogramming paradigms as shown in Figure 2:

---

**Algorithm 1** Hybrid Architecture

**Require:** $\{x_d\}_{d=1}^{D}, \{a_d\}_{d=1}^{D}, \lambda$.

Initialize $z_{NRPM}^{(0)} = z_{LPM} = \sum_{d=1}^{D} a_d \cdot x_d$

**for** $k = 0, 1, \ldots, K - 1$ **do**

$\quad w_d^{(k)} = \frac{1}{|a_d x_d - z_{NRPM}^{(k)}/D|} \forall d \in [D]$

$\quad z_{NRPM}^{(k+1)} = D \cdot \frac{\sum_{d=1}^{D} w_d^{(k)} a_d x_d}{\sum_{d=1}^{D} w_d^{(k)}}$

**end for**

**return** $\lambda \cdot z_{LPM} + (1 - \lambda) \cdot z_{NRPM}^{(K)}$

---

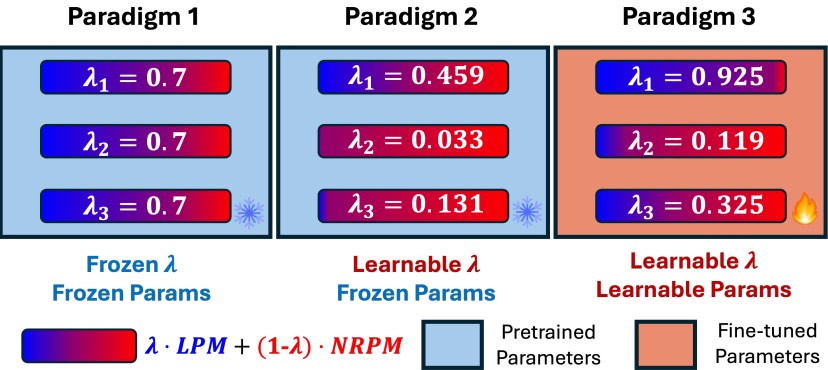

Figure 2: *Three Robustness Reprogramming Paradigms* : (1) Paradigm 1 freezes the model parameters and treats $\{\lambda_l\}_{l=1}^{L}$ as fixed hyperparameter; (2) Paradigm 2 freezes the model parameters but allows $\{\lambda_l\}_{l=1}^{L}$ to be learnable; (3) Paradigm 3 enables both the model parameters and $\{\lambda_l\}_{l=1}^{L}$ to be learnable.

**Paradigm 1: without fine-tuning, good robustness with zero cost.** As deep learning models become increasingly larger, it is critical to fully utilize pre-trained model parameters. Since the robust NRPM slightly refines the vanilla LPM through an adaptive instance-wise reweighting scheme, ensuring that the pre-trained LPM parameters still fit well within NRPM architecture. Additionally, by adjusting the hyperparameters $\{\lambda_l\}_{l=1}^{L}$ with pre-trained parameters, we can strike an ideal balance between natural and robust accuracy. It is worth noting that relying solely on pre-trained parameters with plug-and play paradigm significantly reduces computational costs, which is crucial in the era of large-scale deep learning.

**Paradigm 2: only fine-tuning** $\{\lambda_l\}_{l=1}^{L}$**, strong robustness with slight cost.** One drawback of Paradigm 1 is that we specify the same $\lambda$ for all the layers and need to conduct brute force hyperparameters search to obtain the optimal one. However, hyper-parameters search is time-consuming and computation-intensive. Moreover, the entire model requires layer-wise $\{\lambda_l\}_{l=1}^{L}$ to balance the LPM and NRPM for different layers. To solve it, we propose Paradigm 2, which automatically learns optimal $\lambda$ with light-weight fine-tuning. This paradigm only fine-tunes the hyperparameters $\{\lambda_l\}_{l=1}^{L}$ while keeping the model parameters frozen, which is very efficient.

**Paradigm 3: overall fine-tuning, superior robustness with acceptable cost.** To achieve best robustness, we can make both the model parameters $\{a_d\}_{d=1}^{D}$ and the hyperparameters $\{\lambda_l\}_{l=1}^{L}$ learnable. By refining these parameters based on the pre-trained model, we can prevent clean performance degradation. Moreover, with learnable $\{\lambda_l\}_{l=1}^{L}$ during adversarial training, the model can automatically select the optimal combination to enhance both clean and robust performance. This paradigm is still efficient since it only needs to fine-tune the model lightly with a few training epochs.

## 5 EXPERIMENT

In this section, we comprehensively evaluate the effectiveness of our proposed *Robustness Reprogramming* techniques using a wide range of backbone architectures, starting from basic MLPs, progressing to shallow LeNet model, and extending to the widely-used ResNets architecture.

### 5.1 EXPERIMENTAL SETTING

**Datasets.** We conduct the experiments on MNIST LeCun & Cortes (2005), SVHN (Netzer et al., 2011), CIFAR10 (Krizhevsky et al., 2009), and ImageNet10 (Russakovsky et al., 2015) datasets.

**Backbone architectures.** We select backbones ranging from very basic MLPs with 1, 2, or 3 layers, to mid-level architectures like LeNet, and deeper networks such as ResNet10, ResNet18, and ResNet34 (He et al., 2016). In some experiments with ResNets, we chose the narrower version (with the model width reduced by a factor of 8) for the consideration for computation issue. Additionally, we also choose one popular architecture MLP-Mixer (Tolstikhin et al., 2021).

**Evaluation methods.** We assess the performance of the models against various attacks under $L_\infty$ norm, including FGSM (Goodfellow et al., 2014), PGD-20 (Madry, 2017), C&W (Carlini & Wagner, 2017), and AutoAttack (Croce & Hein, 2020b). Among them, AutoAttack is an ensemble attack consisting of three adaptive white-box attacks and one black-box attack, which is considered as a reliable evaluation method to avoid the false sense of security. In addition to empirical robustness evaluation, we also evaluate certified robustness to further demonstrate the robustness of our proposed architecture.

**Baselines & Hyperparameter setting.** For backbone ResNets, we compare the baselines including PGD-AT (Madry, 2017), TRADES (Zhang et al., 2019), MART (Wang et al., 2019), SAT (Huang et al., 2020), and AWP (Wu et al., 2020). We train the baselines for 200 epochs with batch size 128, weight decay 2e-5, momentum 0.9, and an initial learning rate of 0.1 that is divided by 10 at the 100-th and 150-th epoch. For the backbone MLPs and LeNet, we train the vanilla models for 50 epochs. Our robustness reprogramming will fine-tune the pre-trained models for 5 epochs.

## 5.2 ROBUSTNESS REPROGRAMMING ON MLPs

Table 1: Robustness reprogramming under 3 paradigms on MNIST with 3-layer MLP as backbone.

| Method / Budget | Natural | 0.05 | 0.1 | 0.15 | 0.2 | 0.25 | 0.3 | $[\lambda_1, \lambda_2, \lambda_3]$ |
|---|---|---|---|---|---|---|---|---|
| Normal-train | **90.8** | 31.8 | 2.6 | 0.0 | 0.0 | 0.0 | 0.0 | \ |
| Adv-train | 76.4 | 66.0 | 57.6 | 46.9 | 35.0 | 23.0 | 9.1 | \ |
| | **90.8** | 31.8 | 2.6 | 0.0 | 0.0 | 0.0 | 0.0 | [1.0, 1.0, 1.0] |
| | **90.8** | 56.6 | 17.9 | 8.5 | 4.6 | 3.0 | 2.3 | [0.9, 0.9, 0.9] |
| | 90.4 | 67.1 | 30.8 | 17.4 | 10.6 | 6.5 | 4.5 | [0.8, 0.8, 0.8] |
| | 89.7 | 73.7 | 43.5 | 25.5 | 16.9 | 11.7 | 9.2 | [0.7, 0.7, 0.7] |
| Paradigm 1 | 88.1 | 75.3 | 49.0 | 31.0 | 22.0 | 15.5 | 12.4 | [0.6, 0.6, 0.6] |
| (without tuning) | 84.1 | 74.4 | 50.0 | 31.9 | 22.8 | 18.1 | 14.3 | [0.5, 0.5, 0.5] |
| | 78.8 | 70.4 | 48.3 | 33.9 | 24.1 | 18.4 | 14.6 | [0.4, 0.4, 0.4] |
| | 69.5 | 62.6 | 45.2 | 31.5 | 23.1 | 19.0 | 15.5 | [0.3, 0.3, 0.3] |
| | 58.5 | 53.2 | 38.2 | 27.6 | 22.2 | 16.4 | 12.9 | [0.2, 0.2, 0.2] |
| | 40.7 | 38.3 | 29.7 | 22.8 | 16.8 | 12.9 | 11.1 | [0.1, 0.1, 0.1] |
| | 18.8 | 17.6 | 16.4 | 14.6 | 12.4 | 10.7 | 9.4 | [0.0, 0.0, 0.0] |
| Paradigm 2 (tuning $\lambda$) | 81.5 | 75.3 | 61.2 | 44.7 | 33.7 | 26.0 | 20.1 | [0.459, 0.033, 0.131] |
| Paradigm 3 (tuning all) | 86.1 | **81.7** | **75.8** | **66.7** | **58.7** | **50.1** | **39.8** | [0.925, 0.119, 0.325] |

**Comparison of robustness reprogramming via various paradigms.** To compare the robustness of our robustness reprogramming under 3 paradigms as well as the vanilla normal/adversarial training, we evaluate the model performance under FGSM attack across various budgets with 3-layer MLP as backbone model. From the results in Table 1, we can make the following observations:

- In terms of robustness, our Robustness Reprogramming across three paradigms progressively enhances performance. In Paradigm 1, by adjusting $\{\lambda_l\}_{l=1}^L$, an optimal balance between clean and robust accuracy can be achieved without the need for parameter fine-tuning. Moreover, Paradigm 2 can automatically learn the layer-wise set $\{\lambda_l\}_{l=1}^L$, improving robustness compared to Paradigm 1 (with fixed $\{\lambda_l\}_{l=1}^L$). Furthermore, Paradigm 3, by fine-tuning all the parameters, demonstrates the best performance among all the methods compared.

- Regarding natural accuracy, we can observe from Paradigm 1 that increasing the inclusion of NRLM (i.e., smaller $\{\lambda_l\}_{l=1}^L$) results in a decline in performance. But this sacrifice can be mitigated by fine-tuning $\{\lambda_l\}_{l=1}^L$ or the entire models as shown in Paradigm 2&3.

**Behavior analysis on automated learning of** $\{\lambda_l\}_{l=1}^L$. In Paradigm 2, we assert and expect that the learnable $\{\lambda_l\}_{l=1}^L$ across layers can achieve optimal performance under a specified noisy environment while maintaining the pre-trained parameters. To further validate our assertion and investigate the behavior of the learned $\{\lambda_l\}_{l=1}^L$, we simulate various noisy environments by introducing adversarial perturbations (FGSM) into the training data at different noise levels, $\epsilon$. We initialize the $\{\lambda_l\}_{l=1}^L$ across all layers to 0.5. From the results shown in Table 2, we can make the following observations:

Table 2: Automated learning of $\{\lambda_l\}_{l=1}^L$ under adversarial training with various noise levels.

| Budget | Natural | 0.05 | 0.1 | 0.15 | 0.2 | 0.25 | 0.3 | Learned $\{\lambda_l\}_{l=1}^L$ |
|---|---|---|---|---|---|---|---|---|
| Adv-train ($\epsilon = 0.0$) | 91.3 | 75.7 | 45.0 | 24.9 | 14.9 | 10.5 | 8.0 | [0.955, 0.706, 0.722] |
| Adv-train ($\epsilon = 0.05$) | 91.2 | 76.8 | 50.6 | 27.8 | 16.6 | 10.5 | 8.1 | [0.953, 0.624, 0.748] |
| Adv-train ($\epsilon = 0.1$) | 91.0 | 82.6 | 62.5 | 41.7 | 26.4 | 19.8 | 14.9 | [0.936, 0.342, 0.700] |
| Adv-train ($\epsilon = 0.15$) | 90.6 | 82.3 | 66.8 | 49.4 | 35.0 | 25.5 | 19.9 | [0.879, 0.148, 0.599] |
| Adv-train ($\epsilon = 0.2$) | 90.3 | 82.5 | 67.5 | 49.3 | 35.8 | 27.0 | 21.3 | [0.724, 0.076, 0.420] |
| Adv-train ($\epsilon = 0.25$) | 87.7 | 81.0 | 66.0 | 48.5 | 36.4 | 26.6 | 21.6 | [0.572, 0.049, 0.243] |
| Adv-train ($\epsilon = 0.3$) | 81.5 | 75.3 | 61.2 | 44.7 | 33.7 | 26.0 | 20.1 | [0.459, 0.033, 0.131] |

- The learned $\{\lambda_l\}_{l=1}^L$ values are layer-specific, indicating that setting the same $\{\lambda_l\}_{l=1}^L$ for each layer in Paradigm 1 is not an optimal strategy. Automated learning of $\{\lambda_l\}_{l=1}^L$ enables the discovery of the optimal combination across layers.

- As the noise level in the training data increases, the learned $\{\lambda_l\}_{l=1}^L$ values tend to decrease, causing the hybrid architecture to resemble a more robust NRPM architecture. This suggests that our proposed Paradigm 2 can adaptively adjust $\{\lambda_l\}_{l=1}^L$ to accommodate noisy environments.

**Ablation studies.** To further investigate the effectiveness of our proposed robust architecture, we provide several ablation studies on backbone size, attack budget measurement, additional backbone MLP-Mixer (Tolstikhin et al., 2021), the number of layers $K$ in the Appendix C.2, Appendix C.3, and Appendix C.4, respectively. These experiments demonstrate the consistent advantages of our method across different backbone sizes and attack budget measurements. Additionally, increasing the number of layers $K$ can further enhance robustness, though at the cost of a slight decrease in clean performance.

### 5.3 ROBUSTNESS REPROGRAMMING ON CONVOLUTION

**Robustness reprogramming on convolution.** We evaluate the performance of our robustness reprogramming with various weight $\{\lambda_l\}_{l=1}^L$ in Figure 3 , where we can observe that incorporating more NRPM-induced embedding will significantly improve the robustness while sacrifice a little bit of clean performance. Moreover, by fine-tuning NRPM-model, we can significantly improve robust performance while compensating for the sacrifice in clean accuracy.

**Adversarial fine-tuning.** Beyond natural training, we also validate the advantage of our NRPM architecture over the vanilla LPM under adversarial training. We utilize the pretrained parameter from LPM architecture, and track the clean/robust performance across 10 epochs under the adversarial fine-tuning for both architectures in Figure 4. The curves presented demonstrate the consistent improvement of NRPM over LPM across all the epochs.

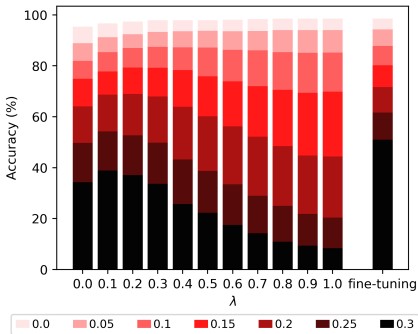

Figure 3: Robustness reprogramming on LeNet. The depth of color represents the size of budget.

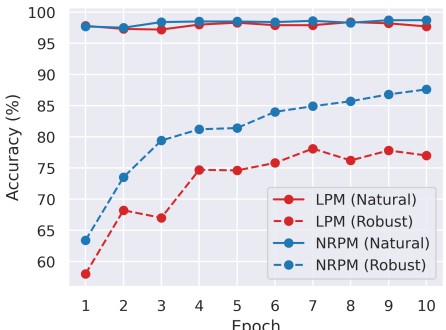

Figure 4: Adversarial fine-tuning on LeNet.

**Hidden embedding visualization.** We also conduct visualization analyses on the hidden embedding to obtain better insight into the effectiveness of our proposed NRPM. First, we quantify the difference between clean embeddings ($x$ or $z_i$) and attacked embeddings ($x'$ or $z_i'$) across all layers in Table 3, and visualize them in Figure 5 and Figure 11. The results in Table 3 show that NRPM-LeNet

has smaller embedding difference across layers, indicating that our proposed NRPM architecture indeed mitigates the impact of the adversarial perturbation. Moreover, as demonstrated in the example in Figure 5, the presence of adversarial perturbations can disrupt the hidden embedding patterns, leading to incorrect predictions in the case of vanilla LeNet. In contrast, our NRPM-LeNet appears to lessen the effects of such perturbations and maintain predicting groundtruth label. From the figures, we can also clearly tell that the difference between clean attacked embeddings of LPM-LeNet is much more significant than in NRPM-LeNet.

Table 3: Embedding difference between clean and adversarial data ($\epsilon = 0.3$) in LeNet. (MNIST)

| LPM-LeNet | $\|\cdot\|_1$ | $\|\cdot\|_2$ | $\|\cdot\|_\infty$ |
|---|---|---|---|
| $|x - x'|$ | 93.21 | 27.54 | 0.30 |
| $|z_1 - z_1'|$ | 271.20 | 116.94 | 1.56 |
| $|z_2 - z_2'|$ | 79.52 | 62.17 | 1.89 |
| $|z_3 - z_3'|$ | 18.77 | 22.56 | 2.32 |
| $|z_4 - z_4'|$ | 9.84 | 18.95 | 3.34 |

| NRPM-LeNet | $\|\cdot\|_1$ | $\|\cdot\|_2$ | $\|\cdot\|_\infty$ |
|---|---|---|---|
| $|x - x'|$ | 90.09 | 26.67 | 0.30 |
| $|z_1 - z_1'|$ | 167.52 | 58.55 | 1.19 |
| $|z_2 - z_2'|$ | 19.76 | 4.27 | 0.56 |
| $|z_3 - z_3'|$ | 3.63 | 0.98 | 0.50 |
| $|z_4 - z_4'|$ | 2.21 | 0.79 | 0.57 |

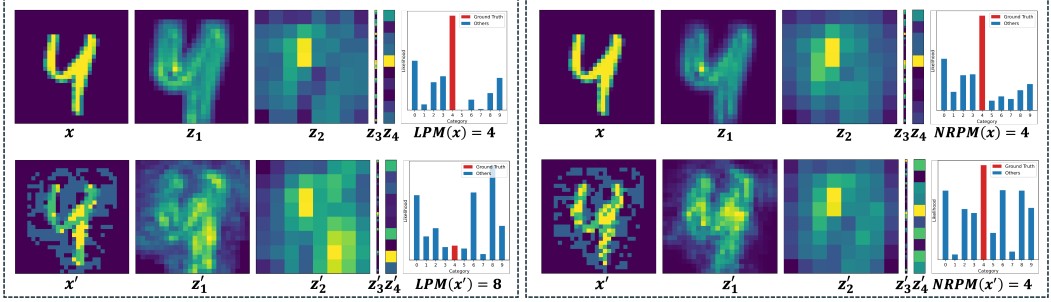

(a) Visualization on LPM-LeNet.  (b) Visualization on NRPM-LeNet.

Figure 5: Visualization of hidden embeddings. The LPM-LeNet is more sensitive to perturbation compared to the NRPM-LeNet: (a) When comparing $z_i$ (1st row) and $z_i'$ (2nd row), LPM (left) shows a more significant difference than NRPM (right). (b) When comparing the likelihood of predictions, the perturbation misleads LPM from predicting 4 to 8, while NRPM consistently predicts 4 in both clean and noisy scenarios.

**Additional experiments.** To further demonstrate the effectiveness of our proposed method, we include experiments on two additional datasets, SVHN and ImageNet10, which are provided in Appendix D.2. All these experiments demonstrate consistent advantages of our proposed method.

## 5.4 ROBUSTNESS REPROGRAMMING ON RESNETS

In this section, we will evaluate the robustness of robustness reprogramming on ResNets across various attacks and further validate the effectiveness under diverse settings in the ablation studies.

Table 4: Robustness reprogramming on CIFAR10 with narrow ResNet18 as backbone.

| Budget $\epsilon$ | Natural | 8/255 | 16/255 | 32/255 | $\{\lambda_l\}_{l=1}^L$ |
|---|---|---|---|---|---|
| | 67.89 | 41.46 | 16.99 | 3.17 | $\lambda = 1.0$ |
| | 59.23 | 40.03 | 23.05 | 8.71 | $\lambda = 0.9$ |
| | 40.79 | 28.32 | 20.27 | 13.70 | $\lambda = 0.8$ |
| Paradigm 1 | 24.69 | 18.34 | 15.31 | 13.09 | $\lambda = 0.7$ |
| (without tuning) | 17.84 | 14.26 | 12.63 | 11.66 | $\lambda = 0.6$ |
| | 15.99 | 12.51 | 11.42 | 11.07 | $\lambda = 0.5$ |
| | 10.03 | 10.02 | 10.0 | 10.0 | $\lambda = 0.4$ |
| Paradigm 2 (tuning $\lambda$) | 69.08 | 44.09 | 24.94 | 12.63 | Learnable |
| Paradigm 3 (tuning all) | 71.79 | 50.89 | 39.58 | 30.03 | Learnable |

**Robustness reprogramming on ResNets.** We evaluate the robustness under PGD of our robustness reprogramming via three paradigms in Table 4. From the results, we can observe that: (1) In Paradigm 1, adding more NRPM-based embeddings without fine-tuning leads to a notable drop in clean performance in ResNet18, which also limits the robustness improvement. (2) In Paradigm 2,

by adjusting only the $\{\lambda_l\}_{l=1}^L$ values, we can improve the clean and robust performance, indicating the need of layer-wise balance across different layers. (3) In Paradigm 3, by tuning both $\{\lambda_l\}_{l=1}^L$ and parameters, we observe that both the accuracy and robustness can be further improved.

**Adversarial robustness.** To validate the effectiveness of our robustness reprogramming with existing method, we select several existing popular adversarial defenses and report the experimental results of backbone ResNet18 under various attacks in Table 5. From the results we can observe that our robustness reprogramming exhibits excellent robustness across various attacks.

Table 5: Adversarial robsustness on CIFAR10 with ResNet18 as backbone.

| Method | Natural | PGD | FGSM | C&W | AA | DeepFool | SPSA | AVG |
|---|---|---|---|---|---|---|---|---|
| PGD-AT | 80.90 | 44.35 | 58.41 | 46.72 | 42.14 | 14.81 | 62.92 | 44.89 |
| TRADES-2.0 | 82.80 | 48.32 | 51.67 | 40.65 | 36.40 | 25.91 | 64.29 | 44.54 |
| TRADES-0.2 | **85.74** | 32.63 | 44.26 | 26.70 | 19.00 | 12.98 | 57.79 | 32.23 |
| MART | 79.03 | 48.90 | 60.86 | 45.92 | 43.88 | 25.63 | 56.55 | 46.96 |
| SAT | 63.28 | 43.57 | 50.13 | 47.47 | 39.72 | 22.34 | 53.47 | 42.78 |
| AWP | 81.20 | 51.60 | 55.30 | 48.00 | 46.90 | 26.25 | 61.37 | 48.24 |
| Consistency | 84.37 | 45.19 | 53.84 | 43.75 | 40.88 | 21.27 | 65.91 | 45.14 |
| DYNAT | 82.34 | 52.25 | 65.96 | 52.19 | 45.10 | 28.72 | **67.97** | 52.03 |
| Paradigm 3 (Ours) | 80.43 | **57.23** | **70.23** | **64.07** | **52.60** | **36.50** | 67.56 | **58.03** |

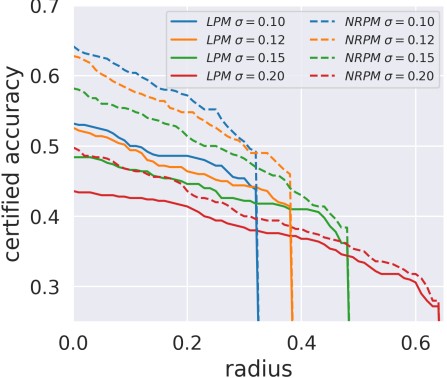

Figure 6: Certified robustness via randomized smoothing with various $\sigma$ levels.

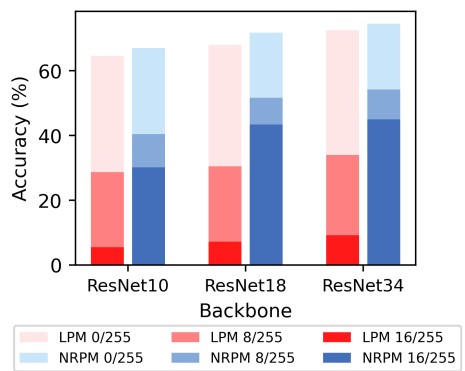

Figure 7: Ablation study on backbone size. The depth of color represents budget size.

**Certified Robustness.** Additionally, we also evaluate the certified robustness using randomized smoothing with various $\sigma$ levels in Figure 6. The curves presented in the figure demonstrate a significant advantage of our NRPM over vanilla LPM, further validating the effectiveness of our proposed architecture.

**Different backbone sizes & budgets.** Here, we conduct ablation studies on the backbone size & budget under AutoAttack in Figure 7 and leave the results under PGD in Appendix E. The results show the evident advantage of our NRPM architecture.

## 6 CONCLUSION

This paper addresses a fundamental challenge in representation learning: how to reprogram a well-trained model to enhance its robustness without altering its parameters. We begin by revisiting the essential linear pattern matching in representation learning and then introduce an alternative non-linear robust pattern matching mechanism. Additionally, we present a novel and efficient Robustness Reprogramming framework, which can be flexibly applied under three paradigms, making it suitable for practical scenarios. Our theoretical analysis and comprehensive empirical evaluation demonstrate significant and consistent performance improvements. This research offers a promising and complementary approach to strengthening adversarial defenses in deep learning, significantly contributing to the development of more resilient AI systems.

STATEMENT

**Ethics statement.** This paper proposes robustness reprogramming techniques to enhance the robustness and safety of machine learning models. We do not identify any potential negative concerns.
**Reproducibility statement.** This paper provides all necessary technique details for reproducibility, including theoretical analysis, algorithm details, experimental settings, pseudo code, implementation, and source code of the proposed techniques.

ACKNOWLEDGMENT

Zhichao Hou and Dr. Xiaorui Liu are supported by the NSF National AI Research Resource Pilot Award, Amazon Research Award, NCSU Data Science Academy Seed Grant Award, and NCSU Faculty Research and Professional Development Award.

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

## A  EFFICIENT IMPLEMENTATION OF ROBUST CONVOLUTION/MLPS

In the main paper, we formulate the pixel-wise algorithm for notation simplicity. However, it is not trivial to implement such theory into practice in convolution and MLPs layer. Here, we will present the detailed implementation or techniques for each case.

### A.1  ROBUST MLPS

In MLPs, we denote the input as $\mathbf{X} \in \mathbb{R}^{D_1}$, the output embedding as $\mathbf{Z} \in \mathbb{R}^{D_2}$, the model parameters as $\mathbf{A} \in \mathbb{R}^{D_1 \times D_2}$. The pseudo code is presented in Algorithm 2.

---

**Algorithm 2** Robust MLPs with NRPM

```
1 def RobustMLP(X, A, eps = 1e-3, K = 3):
2     AX = X.unsqueeze(-1) * A
3     D = X.shape[0]
4     Z = torch.matmul(X, A) # Initialization as LPM-estimation
5     For _ in range(K):
6         DIST = torch.abs(KX - Z/D)  # Distance
7         W = 1/(DIST + eps)
8         W = normalize(W, p=1, dim=0)
9         Z = D*(W*AX).sum(dim=0) # Update
10    return Z
```

---

### A.2  ROBUST CONVOLUTION

**Unfolding.** To apply robust estimation over each patch, we must first unfold the convolution into multiple patches and process them individually. While it's possible to use a for loop to extract patches across the channels, height, and width, this approach is not efficient. Instead, we can leverage PyTorch's `torch.nn.functional.unfold` function to streamline and accelerate both the unfolding and computation process.

## B  THEORETICAL PROOF

### B.1  PROOF OF LEMMA 3.1

*Proof.* Since $\sqrt{a} \leq \frac{a}{2\sqrt{b}} + \frac{\sqrt{b}}{2}$ and the equlity holds when $a = b$, by replacemnet as $a = (a_d x_d - z/D)^2$ and $b = (a_d x_d - z_0/D)^2$, then

$$|a_d x_d - z/D| \leq \frac{1}{2} \cdot \frac{1}{|a_d x_d - z_0/D|} \cdot (a_d x_d - z/D)^2 + \frac{1}{2}|a_d x_d - z_0/D|$$

$$= w_d \cdot (a_d x_d - z/D)^2 + \frac{1}{2}|a_d x_d - z_0/D|$$

Sum up the items on both sides, we obtain

$$\mathcal{L}(z) = \sum_{d=1}^{D} |a_d x_d - z/D| \leq \sum_{d=1}^{D} w_d \cdot (a_d x_d - z/D)^2 + \frac{1}{2}\sum_{d=1}^{D} |a_d x_d - z_0/D| = \mathcal{U}(z, z_0)$$

and the equality holds at $a = b$ ($z = z_0$):

$$\mathcal{U}(z_0, z_0) = \mathcal{L}(z_0). \tag{4}$$

□

### B.2  DERIVATION OF IRLS ALGORITHM

*Proof.*

$$\mathcal{U}'(z^{(k)}, z^{(k)}) = \frac{1}{D}\sum_{d=1}^{D} w_d^{(k)} \cdot 2(z^{(k)}/D - a_d x_d)$$

$$\mathcal{U}''(z^{(k)}, z^{(k)}) = \frac{2}{D^2} \sum_{d=1}^{D} w_d^{(k)}$$

$$z^{(k+1)} = z^{(k)} - \frac{\mathcal{U}'(z^{(k)}, z^{(k)})}{\mathcal{U}''(z^{(k)}, z^{(k)})} \tag{5}$$

$$= z^{(k)} - \frac{\frac{1}{D} \sum_{d=1}^{D} w_d^{(k)} \cdot 2(z^{(k)}/D - a_d x_d)}{\frac{2}{D^2} \sum_{d=1}^{D} w_d^{(k)}} \tag{6}$$

$$= D \cdot \frac{\sum_{d=1}^{D} w_d^{(k)} a_d x_d}{\sum_{d=1}^{D} w_d^{(k)}} \tag{7}$$

where $w_d^{(k)} = \frac{1}{2} \cdot \frac{1}{|a_d x_d - z^{(k)}/D|}$. Since the constant in $w_d^{(k)}$ can be canceled in the updated formulation, we have:

$$w_d^{(k)} = \frac{1}{|a_d x_d - z^{(k)}/D|}.$$

$\square$

### B.3 PROOF OF THEOREM 3.2

*Proof.* For notation simplicity, we denote $y = \frac{1}{D}T_{LPM}(F)$, $y_\epsilon = \frac{1}{D}T_{LPM}(F_\epsilon)$, $z = \frac{1}{D}T_{NRPM}(F)$, $z_\epsilon = \frac{1}{D}T_{NRPM}(F_\epsilon)$. Since $\epsilon$ is very small and $D$ is large enough, we can assume $|a_d x_d - y| \approx |a_d x_d - y_\epsilon|$ for simplicity.

**Influence function for LPM.** The new weighted average $y_\epsilon$ under contamination becomes:

$$y_\epsilon = (1 - \epsilon)y + \epsilon \Delta x$$

Taking the limit as $\epsilon \to 0$, we get:

$$IF(\Delta x; T_{LPM}, F) = \lim_{\epsilon \to 0} \frac{D(y_\epsilon - y)}{\epsilon} = \lim_{\epsilon \to 0} \frac{D\epsilon(\Delta x - y)}{\epsilon} = D(\Delta x - y) = D(\Delta x - z_{LPM}/D)$$

This shows that the weighted average $y$ is directly influenced by $\Delta x$, making it sensitive to outliers.

**Influence function for NRPM.** Next, we consider the reweighted estimate $z$, when we introduce a small contamination at $\Delta x$, which changes the distribution slightly. The contaminated reweighted estimate $z_\epsilon$ becomes:

$$z_\epsilon = \frac{(1 - \epsilon) \sum_{d=1}^{D} w_d a_d x_d + \epsilon w_{\Delta x} \Delta x}{(1 - \epsilon) \sum_{d=1}^{D} w_d + \epsilon w_{\Delta x}}$$

We can simplify the expression for $z_\epsilon$ as:

$$z_\epsilon = \frac{\sum_{d=1}^{D} w_d a_d x_d + \epsilon(w_{\Delta x} \Delta x - \sum_{d=1}^{D} w_d a_d x_d)}{(1 - \epsilon) \sum_{d=1}^{D} w_d + \epsilon w_{\Delta x}}$$

First, expand the difference $z_\epsilon - z$:

$$z_\epsilon - z = \frac{\sum_{d=1}^{D} w_d a_d x_d + \epsilon(w_{\Delta x}\Delta x - \sum_{d=1}^{D} w_d a_d x_d)}{(1-\epsilon)\sum_{d=1}^{D} w_d + \epsilon w_{\Delta x}} - \frac{\sum_{d=1}^{D} w_d a_d x_d}{\sum_{d=1}^{D} w_d}$$

$$= \frac{\sum_{d=1}^{D} w_d a_d x_d \cdot \sum_{d=1}^{D} w_d + \epsilon(w_{\Delta x}\Delta x - \sum_{d=1}^{D} w_d a_d x_d) \cdot \sum_{d=1}^{D} w_d}{[(1-\epsilon)\sum_{d=1}^{D} w_d + \epsilon w_{\Delta x}] \cdot \sum_{d=1}^{D} w_d}$$

$$+ \frac{[(1-\epsilon)\sum_{d=1}^{D} w_d + \epsilon w_{\Delta x}] \cdot \sum_{d=1}^{D} w_d a_d x_d}{[(1-\epsilon)\sum_{d=1}^{D} w_d + \epsilon w_{\Delta x}] \cdot \sum_{d=1}^{D} w_d}$$

$$= \frac{\epsilon(w_{\Delta x}\Delta x - \sum_{d=1}^{D} w_d a_d x_d) \cdot \sum_{d=1}^{D} w_d - \epsilon(w_{\Delta x} - \sum_{d=1}^{D} w_d) \cdot \sum_{d=1}^{D} w_d a_d x_d}{[(1-\epsilon)\sum_{d=1}^{D} w_d + \epsilon w_{\Delta x}] \cdot \sum_{d=1}^{D} w_d}$$

$$= \frac{\epsilon(w_{\Delta x}\Delta x \cdot \sum_{d=1}^{D} w_d - w_{\Delta x} \cdot \sum_{d=1}^{D} w_d a_d x_d)}{[(1-\epsilon)\sum_{d=1}^{D} w_d + \epsilon w_{\Delta x}] \cdot \sum_{d=1}^{D} w_d}$$

Finally, divide the difference by $\epsilon$ and take the limit as $\epsilon \to 0$:

$$IF(\Delta x; T_{NRPM}, F) = \lim_{\epsilon \to 0} \frac{D(z_\epsilon - z)}{\epsilon}$$

$$= D \cdot \frac{w_{\Delta x}\Delta x \cdot \sum_{d=1}^{D} w_d - w_{\Delta x} \cdot \sum_{d=1}^{D} w_d a_d x_d}{\sum_{d=1}^{D} w_d \cdot \sum_{d=1}^{D} w_d}$$

$$= \frac{D w_{\Delta x}\left(\Delta x - z_{NRPM}/D\right)}{\sum_{d=1}^{D} w_d}$$

Since $w_{\Delta x}$ is small for outliers (large $|\Delta x - z_{LPM}/D|$), the influence of $\Delta x$ on NRPM is diminished compared to the influence on LPM. Therefore, NRPM is more robust than LPM because the influence of outliers is reduced.

$\square$

## C  ADDITIONAL EXPERIMENTAL RESULTS ON MLPs

### C.1  MLP-MIXER

MLP-Mixer (Tolstikhin et al., 2021) serves as an alternative model to CNNs in computer vision. Building on the excellent performance of basic MLPs, we further explore the effectiveness of our NRPM with MLP-Mixer, presenting the results in Figure 8 and Table 6. The results validate the effectiveness of our proposed methdod in MLP-Mixer with different blocks across various budgets.

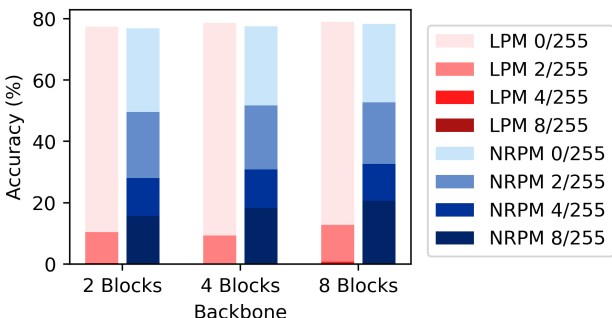

Figure 8: Robustness under PGD on MLP-Mixer. The depth of the color represents the size of the budget.

Table 6: Adversarial Robustness of MLP-Mixer - CIFAR10

| Model | Natural | $\epsilon = 1/255$ | $\epsilon = 2/255$ | $\epsilon = 3/255$ | $\epsilon = 4/255$ | $\epsilon = 8/255$ |
|---|---|---|---|---|---|---|
| LPM-MLP-Mixer-2 | 77.45 | 38.55 | 10.39 | 2.24 | 0.34 | 0.0 |
| NRPM-MLP-Mixer-2 | 76.86 | 64.27 | 49.62 | 37.22 | 27.99 | 15.73 |
| LPM-MLP-Mixer-4 | 78.61 | 35.99 | 9.35 | 1.45 | 0.19 | 0.0 |
| NRPM-MLP-Mixer-4 | 77.51 | 66.72 | 51.73 | 39.43 | 30.88 | 18.31 |
| LPM-MLP-Mixer-8 | 78.98 | 40.55 | 12.81 | 3.27 | 0.73 | 0.01 |
| NRPM-MLP-Mixer-8 | 78.32 | 66.82 | 52.73 | 40.37 | 32.66 | 20.61 |

### C.2  THE EFFECT OF BACKBONE SIZE.

Table 7: Robustness of MLPs with different layers on MNIST.

| Model Size | Arch / Budget | 0 | 0.05 | 0.1 | 0.15 | 0.2 | 0.25 | 0.3 |
|---|---|---|---|---|---|---|---|---|
| 1 Layer | LPM | 91.1 | 12.0 | 7.1 | 0.5 | 0.0 | 0.0 | 0.0 |
| | NRPM | 87.6 | 14.0 | 13.6 | 12.9 | 12.1 | 11.5 | 10.3 |
| 2 Layers | LPM | 91.6 | 32.9 | 2.6 | 0.2 | 0.0 | 0.0 | 0.0 |
| | NRPM | 89.1 | 37.0 | 34.2 | 30.7 | 25.8 | 22.7 | 21.2 |
| 3 Layers | LPM | 90.8 | 31.8 | 2.6 | 0.0 | 0.0 | 0.0 | 0.0 |
| | NRPM | 90.7 | 47.2 | 45.1 | 37.8 | 30.8 | 24.9 | 21.1 |

We select the MLP backbones with different layers to investigate the performance under different model sizes. We report the performance with the linear models of 1 (784 - 10), 2 (784 - 64 -10), 3 (784 - 256 - 64 -10) layers. We present the fine-tuning results for both the LPM model and NRPM model in Table 7, from which we can make the following observations:

• Our RosNet show better robustness with different size of backbones

• Regarding clean performance, linear models with varying layers are equivalent and comparable. However, our RosNet shows some improvement as the number of layers increases.

• The robustness of our RosNet enhances with the addition of more layers. This suggests that our robust layers effectively mitigate the impact of perturbations as they propagate through the network.

## C.3 ATTACK BUDGET MEASUREMENT

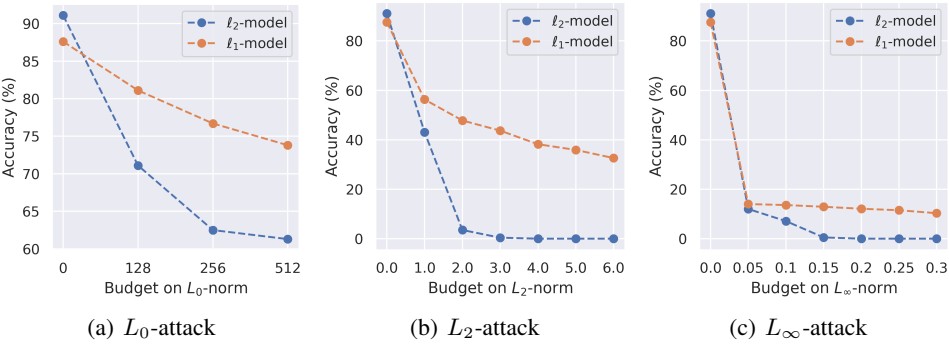

(a) $L_0$-attack  (b) $L_2$-attack  (c) $L_\infty$-attack

Figure 9: Robustness under attacks with different norms.

In previous results of different backbone sizes, we notice the NRPM model is not effective enough under $L_\infty$-attack. To verify the effect the attack budget measurement, we evaluate the robustness under $L_\infty$,$L_2$,$L_0$ attacks where the budget is measured under the $L_\infty$,$L_2$,$L_0$-norms, that is, $\|x - x'\|_p \leq$ budget, where $p = 0, 2, \infty$. We can make the following observations from the results in Figure 9:

- Our NRPM architecture consistently improve the robustness over the LPM-backbone across various attacks and budgets.

- Our NRPM model experiences a slight decrease in clean accuracy, suggesting that the median retains less information than the mean.

- Our NRPM model shows improved performance under the $L_0$ and $L_2$ attacks compared to the $L_\infty$ attack. This behavior aligns with the properties of mean and median. Specifically, under the $L_\infty$ attack, each pixel can be perturbed up to a certain limit, allowing both the mean and the median to be easily altered when all pixels are modified. Conversely, under the $L_2$ and $L_0$ attacks, the number of perturbed pixels is restricted, making the median more resilient to disruption than the mean.

## C.4 EFFECT OF DIFFERENT LAYERS $K$

To investigate the effect of the number of iterations $K$ in the unrolled architecture, we present the results with increasing $K$ on combined models in the Table 8 and Figure 10 in the appendix. As $K$ increases, the hybrid model can approach to the NRPM architecture, which makes the model more robust while slightly sacrificing a little bit natural accuracy.

Table 8: Ablation study on number of layers $K$ - MLP - MNIST

| $K$ / $\epsilon$ | 0 | 0.05 | 0.1 | 0.15 | 0.2 | 0.25 | 0.3 |
|---|---|---|---|---|---|---|---|
| | | | $\lambda = 0.9$ | | | | |
| $K = 0$ | 90.8 | 31.8 | 2.6 | 0.0 | 0.0 | 0.0 | 0.0 |
| $K = 1$ | 90.8 | 56.6 | 17.9 | 8.5 | 4.6 | 3.0 | 2.3 |
| $K = 2$ | 90.7 | 76.1 | 44.0 | 26.8 | 18.0 | 13.4 | 9.3 |
| $K = 3$ | 90.7 | 82.1 | 56.7 | 37.4 | 26.3 | 19.9 | 16.3 |
| $K = 4$ | 90.6 | 83.5 | 61.1 | 42.7 | 29.4 | 22.7 | 18.0 |
| | | | $\lambda = 0.8$ | | | | |
| $K = 0$ | 90.8 | 31.8 | 2.6 | 0.0 | 0.0 | 0.0 | 0.0 |
| $K = 1$ | 90.4 | 67.1 | 30.8 | 17.4 | 10.6 | 6.5 | 4.5 |
| $K = 2$ | 90.4 | 81.1 | 56.2 | 36.2 | 26.0 | 19.3 | 15.5 |
| $K = 3$ | 90.3 | 81.4 | 61.8 | 42.9 | 31.8 | 23.7 | 19.5 |
| $K = 4$ | 90.3 | 82.5 | 64.3 | 45.6 | 33.1 | 25.3 | 22.0 |
| | | | $\lambda = 0.7$ | | | | |
| $K = 0$ | 90.8 | 31.8 | 2.6 | 0.0 | 0.0 | 0.0 | 0.0 |
| $K = 1$ | 89.7 | 73.7 | 43.5 | 25.5 | 16.9 | 11.7 | 9.2 |
| $K = 2$ | 89.1 | 80.3 | 56.4 | 39.7 | 28.4 | 21.4 | 18.5 |
| $K = 3$ | 88.9 | 80.3 | 61.8 | 44.0 | 33.2 | 24.4 | 19.2 |
| $K = 4$ | 88.4 | 79.6 | 61.3 | 43.6 | 33.9 | 26.3 | 21.4 |
| | | | $\lambda = 0.6$ | | | | |
| $K = 0$ | 90.8 | 31.8 | 2.6 | 0.0 | 0.0 | 0.0 | 0.0 |
| $K = 1$ | 88.1 | 75.3 | 49.0 | 31.0 | 22.0 | 15.5 | 12.4 |
| $K = 2$ | 86.2 | 77.1 | 56.2 | 39.3 | 29.2 | 22.6 | 17.1 |
| $K = 3$ | 85.5 | 75.9 | 55.8 | 40.2 | 30.7 | 23.1 | 18.8 |
| $K = 4$ | 84.0 | 74.7 | 56.2 | 39.6 | 29.4 | 22.1 | 18.7 |
| | | | $\lambda = 0.5$ | | | | |
| $K = 0$ | 90.8 | 31.8 | 2.6 | 0.0 | 0.0 | 0.0 | 0.0 |
| $K = 1$ | 84.1 | 74.4 | 50.0 | 31.9 | 22.8 | 18.1 | 14.3 |
| $K = 2$ | 80.3 | 71.0 | 53.5 | 37.0 | 26.9 | 21.5 | 16.4 |
| $K = 3$ | 79.0 | 69.6 | 53.7 | 38.1 | 29.2 | 23.2 | 19.5 |
| $K = 4$ | 77.7 | 66.6 | 51.2 | 37.9 | 29.7 | 23.7 | 19.8 |

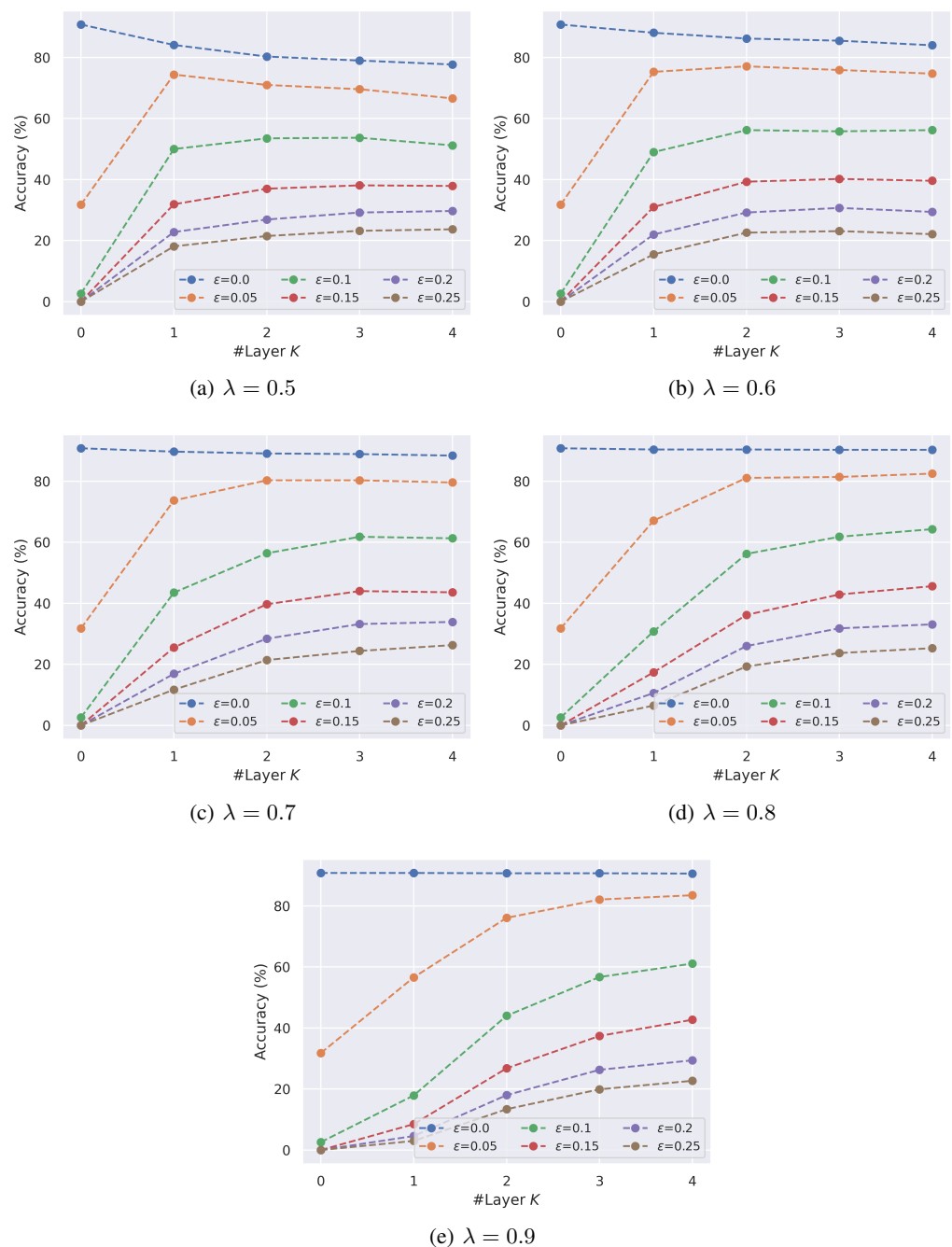

Figure 10: Effect of layers $K$

# D ADDITIONAL EXPERIMENTAL RESULTS ON LENET

## D.1 VISUALIZATION OF HIDDEN EMBEDDING

We put several examples of the visualizations of hidden embedding in Figure 11.

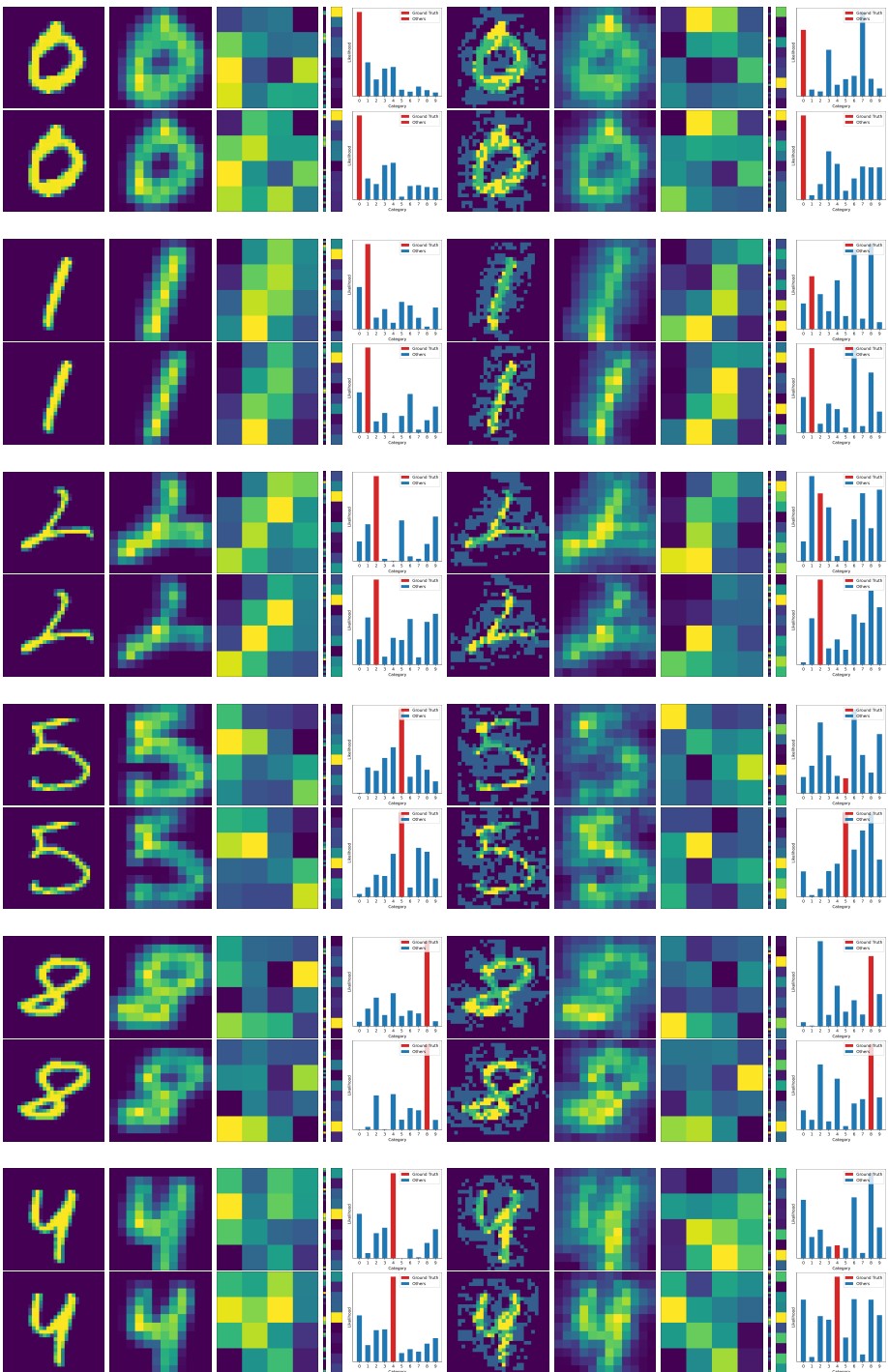

Figure 11: Visualization of hidden embedding.

## D.2 ADDITIONAL DATASETS

Besides MNIST, we also conduct the experiment on SVHN and ImageNet10, and show the results under PGD in Table 9 and Table 10, respectively.

Table 9: Robustness of LeNet under PGD on SVHN.

| Model / Budget $\epsilon$ | 0/255 | 1/255 | 2/255 | 4/255 | 8/255 | 16/255 | 32/255 | 64/255 |
|---|---|---|---|---|---|---|---|---|
| LPM-LeNet | 83.8 | 68.6 | 49.3 | 27.1 | 8.2 | 3.0 | 1.9 | 1.3 |
| NRPM-LeNet | 83.2 | 72.4 | 54.6 | 35.4 | 20.1 | 14.3 | 11.8 | 7.9 |

Table 10: Robustness of LeNet under PGD on ImageNet10.

| Model / Budget $\epsilon$ | 0/255 | 2/255 | 4/255 | 8/255 |
|---|---|---|---|---|
| LPM-LeNet | 53.72 | 13.11 | 7.67 | 3.89 |
| NRPM-LeNet | 53.55 | 15.06 | 13.68 | 12.50 |

# E  ADDITIONAL EXPERIMENTAL RESULTS ON RESNETS

## E.1  EFFECT OF BACKBONE SIZE

We evaluate the robustness under PGD with different backbone sizes including ResNet10, ResNet18, and ResNet34. The results presented in Figure 12 demonstrate the consistent improvement of our NRPM over vanilla LPM.

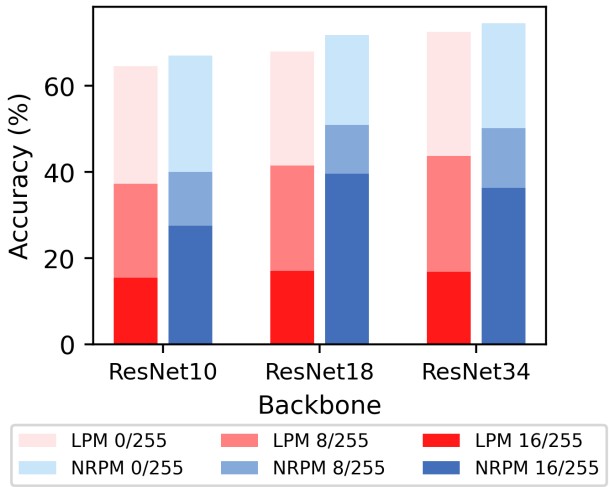

Figure 12: Ablation study on backbone size. (PGD)

