# OpenReview forum: "Robustness Reprogramming for Representation Learning"
_ICLR.cc/2025/Conference — ICLR 2025 Spotlight_

### Official Review · Reviewer_QnFV · 2024-11-03

**Soundness:** 2
**Presentation:** 2
**Contribution:** 2
**Rating:** 8
**Confidence:** 2

**Summary:**

The paper proposes an adversarial robustness scheme that doesn't necessarily require altering the pretrained parameters. They propose a "reprogramming" scheme to adapt the representations learned by a pretrained network in order to enhance the network's robustness. The proposed approach is extensively evaluated on multiple datasets and simpler model architectures and is shown to enhance empirical performance.

**Strengths:**

- The approach is thoroughly evaluated on multiple datasets and simpler models.
- It empirically performs well under increasing attack strength/various attack types.

**Weaknesses:**

1) The readability of the paper is not good in its current form. The paper doesn't provide sufficient background/cite references to understand the terminology used or where ideas are derived from. Overall, I find it very hard to understand the ideas/contributions of this work, even though the results look promising.

2) The notation is inconsistent, and several terms are used without introduction. For example,

      a) What is X in the indicator function of F in line 212?

      b) What is y in Line 223? Such missing details make it very hard to follow the ideas introduced in the paper.

      c) What is the difference between {\lambda} and \lambda? Guessing from Table 1, the weights are learned/assigned per layer? The notation does not convey this correctly (it should either be a vector or otherwise properly subscripted in the notation).

      d) What is N in algorithm 2?

3) The figures lack sufficient description to understand what's happening. How are the visualizations of hidden embeddings created in Figure 5? What is meant by the "size of the budget" in Figure 3? (my intuitive guess is that refers to the attack budget, but under what norm? Which attack? The language needs to be a lot clearer and the captions/descriptions are too succinct and vague in the current draft).

4) Please include an analysis of how exactly is this better than adversarial training. How do the training time and number of parameters compare for the two approaches?

5) I also find it hard to appreciate the need for the 3 separate training paradigms. A discussion of when and how exactly would each of these be more applicable and suitable is lacking.

6) What is the cost of scaling this approach to more practical models/datasets (e.g. ImageNet with ResNet50 or so)? This goes back to -4- as to if/how this is better than AT, besides the empirical gains.

**Questions:**

Refer to weaknesses.

---

> ### Author Response · Authors · 2024-11-22
>
> **W1:** Thanks for pointing out. Our idea of this work comes from the robust statistics and optimization. We are glad to further explain and summarize the idea and contributions here.
>
> The idea logic of Nonlinear Robust Pattern Matching (NRPM):
>
> (a) We first reveal the common module in all the deep learning models, that is, the linear feature transformation, $z = \mathbf{a}^\top \mathbf{x} = \sum_{d=1}^D a_d \cdot x_d$.
>
> (b) The linear feature transformation $z = \mathbf{a}^\top \mathbf{x}$ has a underlying inverse optimization problem, Ordinary Least Squares (OLS) problem:
>
> $$
> \min_{z \in \mathbb{R}} \mathcal{L}(z) = \sum_{d=1}^D \left( \frac{z}{D} - a_d \cdot x_d \right)^2,
> $$
>
> where the first-order optimality condition $\frac{\partial \mathcal{L}(z)}{\partial z} = 0$ yields the optimal solution $z^* = \sum_{d=1}^D a_d \cdot x_d$.
>
> (c) OLS estimation is highly sensitive to outlying values due to the quadratic penalty, so we choose to use linear penalty which give a robust alternative Least Absolute Deviation (LAD) estimation:
>
> $$
> \min_{z \in \mathbb{R}} \mathcal{L}(z) = \sum_{d=1}^D \left| \frac{z}{D} - a_d \cdot x_d \right|.
> $$
>
> (d) LAD estimation is non-smooth, which requires an Newton-IRLS algorithm to approximate the estimator and unroll it as the neural network layer as:
>
> $$
> z^{(k+1)} = D \cdot \frac{\sum_{d=1}^D w_d^{(k)} a_d x_d}{\sum_{d=1}^D w_d^{(k)}}.
> $$
>
> The significant contributions can be found in the common response.
>
>
>
>
>
>
>
> **W2:** Thank you for your comments. We have corrected the typos and provided clearer explanations in the revised version accordingly.
>
> (a)  Consider a feature data points {${a_d x_d}$}$_{d=1}^D$. $X$ represent the random variable of feature empirical distribution $F$.
>
> (b)  $y = z_{LPM}/D$ is defined as in the following formulation after “=>”. This definition is introduced for the sake of notational simplicity in the proof of Theorem 3.2 in the appendix. We will further clarify it in the revised version.
>
> (c)  {${\lambda}$} represents a set of layer-wise hyperparameters. Each layer will have one $\lambda$.
>
> (d) $N$ is a typo and should be $D$, which is the dimension of the input data.
>
>
>
> **W3:** We have included the detailed explanation in the **Hidden Embedding Visualization** part in Section 5.3. And we are glad to explain it again here: In Figure 5, we visualize the clean embeddings ($x$ or $z_i$, first row) and the attacked embeddings ($x'$ or $z_i'$, second row) of LPM (first column) and NRPM (second column). We then compare the clean-attacked embedding differences between LPM and NRPM.
>
> It is evident that under attack, in the vanilla LPM, the hidden embeddings are more seriously destroyed, and the output probability changes significantly, which explains its reduced robustness. We will revise the caption with more detailed explanations following your kind suggestions.

---

> ### Author Response · Authors · 2024-11-22
>
> **W4:** We want to emphasize that our proposed method is orthogonal to the existing methods including adversarial training. It can be combined with adversarial training to further improve the robustness. Our proposed method can be considered as a post-hoc approach for any well-trained models, which reprograms the vanilla architecture into a robust one.
>
> In terms of training time, we pre-train the model with adv-train technique for 200 epochs, after which we fine-tune the hybrid architecture for only 3-5 epochs. They are not directly comparable as they operate at different stages. We also acknowledge that our architecture includes additional computational costs; however, this is not the primary focus of our work. Most importantly, we would like to highlight the significant contributions of our work to the field of robust machine learning as explained in W1.
>
> In terms of the number of parameters, we include only a few additional {${\lambda}$}, which are negligible in large-scale models.
>
>
>
> **W5:** We believe we have explained and discussed the suitable scenarios for each paradigm in Section 4. We are happy to provide further clarification here:
>
> - **Paradigm 1 (without fine-tuning):** is suitable when the model is large-scale and pre-trained, and there is no available training data. In this case, we can simply adjust the hyperparameters {${\lambda}$} to achieve the desired balance between clean and robust performance.
>
> - **Paradigm 2 (fine-tuning ${\lambda}$ only):** is suitable when the large-scale backbone models have substantial parameters, but only limited data is available. In this scenario, we can freeze the model parameters and fine-tune only {${\lambda}$} to obtain the optimal layer-wise combination of the hybrid architecture.
>
> - **Paradigm 3 (overall fine-tuning):** is suitable when excellent robustness is required, and it is acceptable to fine-tune the models for a few epochs.
>
>
>
> **W6:** To provide an overview of the scaling cost, we also include a comparison of the inference running time between the vanilla LPM and our NRPM across various layers in the table below. As shown, our NRPM incurs only 3-4 times the inference cost of the LPM, which we consider acceptable. For the question "if/how this is better than AT", please refer to W4.
>
> **Table 15: Running time with ResNet as backbone.**
>
> | Layers   | 10     | 18     | 26     | 34     | 42     | 50     | 58     | 66     | 74     |
> |----------|--------|--------|--------|--------|--------|--------|--------|--------|--------|
> | LPM (ms) | 27.81  | 47.10  | 66.50  | 87.18  | 108.78 | 125.65 | 130.08 | 142.42 | 159.56 |
> | NRPM (ms)| 88.53  | 167.21 | 253.63 | 338.78 | 416.78 | 497.46 | 552.18 | 628.35 | 732.67 |

---

> ### Comment · Reviewer_QnFV · 2024-11-25
>
> Thank you for clarifying these details and for updating the manuscript with corrections/clarifications. I am raising my score since the contributions of this work are significant and are well-supported by empirical results.

---

> > ### Author Response · Authors · 2024-11-25
> >
> > Thank you for your feedback! We really appreciate your recognition of our significant contributions in this work.

---

### Official Review · Reviewer_kryy · 2024-11-03

**Soundness:** 2
**Presentation:** 3
**Contribution:** 3
**Rating:** 8
**Confidence:** 3

**Summary:**

This paper proposed nonlinear robust pattern matching algorithm (NRPM) that can improves the robustness without modifying its parameters while maintaining the feature pattern matching by reprogramming a pretrained model. Especially, this paper theoretically analyzed linear pattern matching and Newton-iterative reweighted least squares algorithm, and introduced robustness reprogramming paradigms under three different situations: parameter-free tuning, lightweight fine-tuning and full fine-tuning. The experimental results support its theoretical analysis and demonstrate the effectiveness of proposed method.

**Strengths:**

- This paper provides comprehensive and detailed theoretical analysis and explanations.
- This paper is well written to follow.

**Weaknesses:**

- This paper leveraged old baselines for ResNet18 backbone while recent baselines (e.g., [1], [2]) are missing and evaluation against stronger attack (e.g., LGV, SPSA, DeepFool) could be beneficial.
   - [1] Consistency regularization for adversarial robustness
   - [2] Dynamic Label Adversarial Training for Deep Learning Robustness Against Adversarial Attacks
- This paper utilizes backbones that are too small and tasks that are too easy. For example, main table (Table 1, 2) used MLPs and MNIST. It would be beneficial to provide experimental results on CIFAR-100 and Tiny-ImageNet at least with backbone of ResNet18 and WideResNet28-10.
- Under same computing resource, it would be better to directly compare the required resources (e.g., FLOPs) or training time (e.g., GPU Hours) between the proposed method (e.g., paradigm 3, full-tuning) and existing methods for ResNet18 backbone.

**Questions:**

- In table 5, robust accuracy against AutoAttack seems to be high, and it would be better to check whether the model suffers from obfuscated gradient or not. For example, following scenarios could be provided: 1) evaluation against adaptive attacks, 2) checking for gradient masking, or 3) evaluation against PGD attacks with extremely large attack strength where robust accuracy should be almost zero while PGD attacks using same strength but different step sizes and steps (e.g., 2 times smaller step size but 2 times larger attack steps) should achieve similar robust accuracy from original attack setting.
- For the proposed three different paradigms, it would be better to explain when it is more effective to use which paradigm to achieve robustness efficiently based on the differences between three scenarios.

---

> ### Author Response · Authors · 2024-11-22
>
> **W1:** We emphasize that our proposed method is orthogonal to existing methods and can be combined with them to further improve performance. Additionally, PGD, FGSM, C&W, and AutoAttack are commonly used and reliable evaluation methods in this field, as demonstrated in most related papers [1][2][3] including the two you mentioned.
>
> To further validate the effectiveness of our method, we also include the baselines and attacks in our evaluation, as shown in the following table. For DeepFool, we also include the results across various attack iterations. The experiments demonstrate excellent robustness compared to the baselines under various attacks.
>
> References:
> 1. Tack, Jihoon, et al. "Consistency regularization for adversarial robustness." 2022.
> 2. Liu, Zhenyu, et al. "Dynamic Label Adversarial Training for Deep Learning Robustness Against Adversarial Attacks." 2024.
> 3. Wu, Dongxian, Shu-Tao Xia, and Yisen Wang. "Adversarial weight perturbation helps robust generalization." 2020.
>
> **Table 7: Adversarial robustness on CIFAR10 with ResNet18 as backbone.**
>
> | Method         | Natural | PGD   | FGSM  | C&W   | AA    | DeepFool | SPSA  | AVG   |
> |----------------|---------|-------|-------|-------|-------|----------|-------|-------|
> | PGD-AT         | 80.90   | 44.35 | 58.41 | 46.72 | 42.14 | 14.81    | 62.92 | 44.89 |
> | TRADES-2.0     | 82.80   | 48.32 | 51.67 | 40.65 | 36.40 | 25.91    | 64.29 | 44.54 |
> | TRADES-0.2     | **85.74** | 32.63 | 44.26 | 26.70 | 19.00 | 12.98    | 57.79 | 32.23 |
> | MART           | 79.03   | 48.90 | 60.86 | 45.92 | 43.88 | 25.63    | 56.55 | 46.96 |
> | SAT            | 63.28   | 43.57 | 50.13 | 47.47 | 39.72 | 22.34    | 53.47 | 42.78 |
> | AWP            | 81.20   | 51.60 | 55.30 | 48.00 | 46.90 | 26.25    | 61.37 | 48.24 |
> | Consistency    | 84.37   | 45.19 | 53.84 | 43.75 | 40.88 | 21.27    | 65.91 | 45.14 |
> | DYNAT          | 82.34   | 52.25 | 65.96 | 52.19 | 45.10 | 28.72    | **67.97** | 52.03 |
> | Paradigm 3 (Ours) | 80.43 | **57.23** | **70.23** | **64.07** | **52.60** | **36.50** | 67.56 | **58.03** |
>
> **Table 8: Adversarial robustness on CIFAR10 with ResNet18 as backbone.**
>
> | Model \ Iteration      | DeepFool-1 | DeepFool-2 | DeepFool-3 | DeepFool-5 |
> |------------------------|------------|------------|------------|------------|
> | PGD-AT                | 61.79      | 40.32      | 14.81      | 0.82       |
> | TRADES-2.0            | 66.35      | 55.99      | 25.91      | 3.45       |
> | TRADES-0.2            | 54.59      | 37.04      | 12.98      | 1.61       |
> | MART                  | 63.18      | 44.88      | 25.63      | 4.38       |
> | SAT                   | 61.45      | 53.67      | 22.34      | 2.12       |
> | AWP                   | 62.18      | 31.04      | 26.25      | 2.76       |
> | Consistency           | 63.54      | 47.63      | 21.27      | 1.63       |
> | DYNAT                 | 65.22      | 50.01      | 28.72      | 5.38       |
> | Paradigm 3 (Ours)     | 68.75      | **61.99**  | **36.50**  | **10.43**  |

---

> ### Author Response · Authors · 2024-11-22
>
> **W2:** We want to emphasize that the tasks we try are not easy since robustness is a very challenging problem (for instance, robust accuracy on CIFAR10 is not yet high yet), and currently existing works suffer from a clear performance limitation. This work proposes fundamental research to explore new strategies for a transformative breakthrough. The size of the backbone and dataset are not the focus of this paper. Instead, we focus on delving deep into the investigation, even starting from very basic deep learning units such as MLPs and LeNets, hoping to provide clear evidence to demonstrate the advantages of our technical ideas, excluding the impact of irrelevant designs.
>
> Following your suggestions, we have included additional experiments on CIFAR-100, Tiny-ImageNet, and WideResNet28-10, as shown in the following tables. Due to time and computational constraints, we only present the results for Paradigm 1.
>
> From the results, we observe that, even without fine-tuning (Paradigm 2), our method can significantly improve over the vanilla backbone, achieving gains of up to 15.94\%, 19.3\%, and 20.2\% in the experiments on CIFAR-100, Tiny-ImageNet, and WideResNet28-10, respectively.
>
>
> **Table 9: Robustness reprogramming on CIFAR100 with ResNet18 as backbone.**
>
> | Natural | PGD   | $\{\lambda\}$             |
> |---------|-------|---------------------------|
> | 72.76   | 6.98  | $\lambda = 1.0$ (Vanilla) |
> | 72.43   | 22.92 | $\lambda = 0.9$           |
> | 72.09   | 22.26 | $\lambda = 0.8$           |
> | 70.76   | 21.59 | $\lambda = 0.7$           |
> | 67.77   | 19.93 | $\lambda = 0.6$           |
> | 59.80   | 14.95 | $\lambda = 0.5$           |
> | 46.84   | 9.64  | $\lambda = 0.4$           |
> | 27.91   | 7.64  | $\lambda = 0.3$           |
> | 12.62   | 4.65  | $\lambda = 0.2$           |
> | 4.32    | 1.99  | $\lambda = 0.1$           |
> | 1.99    | 1.66  | $\lambda = 0.0$           |
>
> **Table 10: Robustness reprogramming on Tiny-Imagenet with ResNet50 as backbone.**
>
> | Natural (Top1) | PGD (Top1) | Natural (Top5) | PGD (Top5) | $\{\lambda\}$             |
> |----------------|------------|----------------|------------|---------------------------|
> | 63.6           | 1.7        | 82.7           | 8.2        | $\lambda = 1.0$ (Vanilla) |
> | 63.0           | 7.9        | 82.0           | 23.4       | $\lambda = 0.9$           |
> | 62.7           | 10.4       | 82.1           | 26.9       | $\lambda = 0.8$           |
> | 62.8           | 12.5       | 81.8           | 27.5       | $\lambda = 0.7$           |
> | 61.7           | 10.4       | 81.7           | 27.1       | $\lambda = 0.6$           |
>
> **Table 11: Robustness reprogramming on CIFAR10 with WideResNet28-10 as backbone.**
>
> | Natural | PGD   | $\{\lambda\}$             |
> |---------|-------|---------------------------|
> | 97.68   | 14.24 | $\lambda = 1.0$ (Vanilla) |
> | 97.35   | 19.21 | $\lambda = 0.9$           |
> | 94.57   | 19.54 | $\lambda = 0.8$           |
> | 91.07   | 19.54 | $\lambda = 0.7$           |
> | 86.05   | 19.54 | $\lambda = 0.6$           |
> | 91.39   | 34.44 | $\lambda = 0.5$           |
>
>
>
>
>
>
> **W3:** Following your suggestion, we  include a comparison of the inference running time between
> the vanilla LPM and our NRPM in the table below. As shown, our NRPM incurs only 3-4 times
> the inference cost of the LPM, which we consider acceptable.
>
>
> **Table 12: Running time with ResNet as backbone.**
>
> | Layers  | 10     | 18     | 26     | 34     | 42     | 50     | 58     | 66     | 74     |
> |---------|--------|--------|--------|--------|--------|--------|--------|--------|--------|
> | LPM (ms)| 27.81  | 47.10  | 66.50  | 87.18  | 108.78 | 125.65 | 130.08 | 142.42 | 159.56 |
> | NRPM (ms)| 88.53 | 167.21 | 253.63 | 338.78 | 416.78 | 497.46 | 552.18 | 628.35 | 732.67 |
>
>
>
> Additionally, we also report the training time of different methods with the same device. As
> shown in the table, our paradigm 3 requires 2-5 times training time of other methods per epoch.
> however, we can only slightly finetune the pretrained model for only 5 epochs, which largely
> reduces the total training time.
>
>
>
> **Table 13: Training time with ResNet18 as backbone in RTX A6000.**
>
> | Method     | Training time         |
> |------------|-----------------------|
> | PGD-AT     | 3.2 min × 200 epochs |
> | TRADES     | 3.4 min × 200 epochs |
> | AWP        | 6.7 min × 200 epochs |
> | Paradigm 3 | 14.6 min × 5 epochs  |
>
> Last, we want to point out that our main contribution lies in proposing a novel approach to
> improve robustness, which is orthogonal to existing methods. Regarding efficiency, while we
> acknowledge that our proposed architecture introduces additional, yet acceptable, computational
> costs, our robustness reprogramming improves robustness in a lightweight manner, either without
> tuning or with slight tuning.

---

> ### Author Response · Authors · 2024-11-22
>
> **Q1:** We believe our evaluation is already adaptive and comprehensive enough to avoid the issue of gradient masking:
>
> - In our proposed NRPM layer, there is no non-differentiable operator, which mitigates concerns about differentiability issues.
> - Our evaluation using AutoAttack already includes the black-box square attack, which does not rely on gradients.
> - In Section 5.4, we also evaluate the certified robustness using randomized smoothing, which theoretically guarantees robustness under any attack.
> - We include a visualization analysis of the hidden embeddings in Section 5.3 to further validate the robustness of our proposed robust layer.
> - Following your suggestions, we evaluate our model with extremely large attack strength and observe that the robust accuracy is almost zero.
>
> **Table 14: Gradient Masking Test on ResNet18**
>
> | Budget   | 0/255 | 8/255 | 16/255 | 32/255 | 64/255 | 128/255 | 256/255 | 512/255 |
> |----------|-------|-------|--------|--------|--------|---------|---------|---------|
> | Accuracy | 59.23 | 40.03 | 23.05  | 8.71   | 2.54   | 0.98    | 0.0     | 0.0     |
>
>
>
> **Q2:** We believe we have explained and discussed the suitable scenarios for each paradigm in Section 4. We are happy to provide further clarification here:
>
> - **Paradigm 1 (without fine-tuning):** is suitable when the model is large-scale and pre-trained, and there is no available training data. In this case, we can simply adjust the hyperparameters {${\lambda}$} to achieve the desired balance between clean and robust performance.
>
> - **Paradigm 2 (fine-tuning ${\lambda}$ only):** is suitable when the large-scale backbone models have substantial parameters, but only limited data is available. In this scenario, we can freeze the model parameters and fine-tune only {${\lambda}$} to obtain the optimal layer-wise combination of the hybrid architecture.
>
> - **Paradigm 3 (overall fine-tuning):** is suitable when excellent robustness is required, and it is acceptable to fine-tune the models for a few epochs.

---

> > ### Comment · Reviewer_kryy · 2024-11-26
> >
> > Thank you for your detailed responses to my questions. Most of my concerns are resolved.
> > Still, the proposed method has huge clean performance drops, which is also an important problem to consider when proposing a new method. In response, I will slightly increase my score.

---

> > > ### Author Response · Authors · 2024-11-27
> > >
> > > Thank you for your valuable feedback. We believe there is some misunderstanding about the performance of our approach. **In fact, our method does not have a huge clean performance drop, and it already achieves the best results compared with existing works.** We would like to address your remaining concerns on the "huge clean performance drops" as follows.
> > >
> > > First, we believe the "huge clean performance drop" you mentioned originates from **Paradigm 1**, where we only set a single hyperparameter $\lambda$ for all layers without any fine-tuning. However, **Paradigm 1** is a zero-cost approach aimed at efficiently improving robustness, and it is not the optimal case. Therefore, we also provide **Paradigm 2** (tuning layer-wise $\lambda$) and **Paradigm 3** (tuning both $\lambda$ and the model parameters) to enhance both clean and robust performance simultaneously, as illustrated in the following table.
> > >
> > > **Table 1: Robustness reprogramming on CIFAR10 with narrow ResNet18 as backbone. The original model is trained under PGD adversarial training.**
> > >
> > > | Budget $\epsilon$ | Natural | 8/255 | 16/255 | 32/255 | {$\lambda$}$_{\ell=1}^{L}$ |
> > > |--------------------|---------|-------|--------|--------|----------------------------|
> > > | **Original model** | 67.89   | 41.46 | 16.99  | 3.17   | $\lambda = 1.0$           |
> > > |                    | 67.89   | 41.46 | 16.99  | 3.17   | $\lambda = 1.0$           |
> > > |                    | 59.23   | 40.03 | 23.05  | 8.71   | $\lambda = 0.9$           |
> > > |                    | 40.79   | 28.32 | 20.27  | 7.03   | $\lambda = 0.8$           |
> > > | **Paradigm 1**     | 24.69   | 18.34 | 15.31  | 13.10  | $\lambda = 0.7$           |
> > > | (without tuning)   | 17.84   | 14.26 | 12.63  | 11.06  | $\lambda = 0.6$           |
> > > |                    | 15.99   | 12.51 | 11.42  | 11.07  | $\lambda = 0.5$           |
> > > |                    | 10.03   | 10.02 | 10.0   | 10.0   | $\lambda = 0.4$           |
> > > | **Paradigm 2**  (tuning $\lambda$)    | 69.08   | 44.09 | 24.94  | 12.63  | Learnable                 |
> > > | **Paradigm 3**   (tuning all)     | 71.79   | 50.89 | 39.58  | 30.03  | Learnable                 |

---

> ### Author Response · Authors · 2024-11-27
>
> Second, our method already achieves the best balance between clean and robust performance among the existing adversarial defenses. From the results shown in the following table, we can observe the following:
>
> - Compared to SAT and MART, our method significantly improves both clean and robust performance.
>
> - Compared to PGD-AT, TRADES-2.0, AWP, and DYNAT, our method demonstrates a significant advantage in robustness while achieving comparable clean performance.
>
> - Although TRADES-0.2 and Consistency yield better clean performance than our method, their robust performance is significantly inferior. In fact, we can use the hyperparameter in the loss of TRADES to adjust our model fine-tuning to achieve similar clean performance but much better robustness compared with these methods.
>
> From these observations, we can conclude that our method significantly improves robustness while maintaining comparable clean performance compared with existing approaches.
>
> **Table 2: Adversarial robustness on CIFAR10 with ResNet18 as backbone.**
>
> | Method          | Natural | PGD   | FGSM  | C&W   | AA    | DeepFool | SPSA  | AVG   |
> |------------------|---------|-------|-------|-------|-------|----------|-------|-------|
> | PGD-AT          | 80.90   | 44.35 | 58.41 | 46.72 | 42.14 | 14.81    | 62.92 | 44.89 |
> | TRADES-2.0      | 82.80   | 48.32 | 51.67 | 40.65 | 36.40 | 25.91    | 64.29 | 44.54 |
> | TRADES-0.2      | **85.74** | 32.63 | 44.26 | 26.70 | 19.00 | 12.98    | 57.79 | 32.23 |
> | MART            | 79.03   | 48.90 | 60.86 | 45.92 | 40.23 | 26.53    | 56.55 | 46.96 |
> | SAT             | 63.28   | 43.57 | 50.13 | 47.47 | 39.72 | 22.34    | 53.47 | 42.78 |
> | AWP             | 81.20   | 51.60 | 55.33 | 46.88 | 40.85 | 26.91    | 67.92 | 48.64 |
> | Consistency     | 84.37   | 45.19 | 53.84 | 43.75 | 40.88 | 21.27    | 69.51 | 45.14 |
> | DYNAT           | 82.34   | 52.25 | 65.96 | 52.19 | 45.10 | 28.72    | **67.97** | 52.03 |
> | **Paradigm 3 (Ours)** | 80.43   | **57.23** | **70.23** | **64.07** | **52.60** | **36.50** | 67.56 | **58.03** |
>
>
> Third, there is an **inherent trade-off between clean and robust performance** in any model. Compared to standard training, adversarial training will reduce clean performance while enhancing robustness. As demonstrated in Table 4 of TRADES [1], the hyperparameter in the loss of TRADES can be adjusted to balance clean and robust accuracy through model training. Orthogonal to these training strategies, our method can adjust this balance by simply adjusting hyperparameters $\lambda$ in the inference process, which is also a notable contribution. This provides unprecedented opportunity for adaptive inference (without fine-tuning), which is a promising future direction for this field.
>
> We hope this can fully address your concern. Please kindly let us know if any further clarification is needed.
>
> [1] Zhang, Hongyang, et al. "Theoretically principled trade-off between robustness and accuracy." *International conference on machine learning*. PMLR, 2019.

---

> > ### Comment · Reviewer_kryy · 2024-12-03
> >
> > Thank you for pointing out my concerns. I acknowledge the novelty and new direction of robust training along with the performance gain, I raised my score to 8.

---

> > > ### Author Response · Authors · 2024-12-03
> > >
> > > Thank you for your feedback! We sincerely appreciate your recognition and support for our efforts.

---

### Official Review · Reviewer_ktRd · 2024-11-04

**Soundness:** 3
**Presentation:** 3
**Contribution:** 3
**Rating:** 8
**Confidence:** 3

**Summary:**

Many methods have been proposed to improve the adversarial robustness of neural networks, including adversarial training, purification, and regularization. However, these approaches don't take advantage of the learned parameters of models that were not robustly trained. This paper proposed robust reprogramming, which uses those learned parameters to construct a new model that is robust to adversarial perturbations by design. The authors identify linear transformations to essentially be linear pattern matching (LPM), with the pattern being specified by the parameters of the transformation which are capable of matching related features in the input. They further note that linear feature pattern matching is the optimal closed-form solution to the Optimized Least Squares problem, which is particularly sensitive to outliers due to its quadratic penalty. To reduce this sensitivity, this work introduces a new feature pattern matching term based on Least Absolute Deviation estimation, which is called Nonlinear Robust Pattern Matching (NRPM). Theoretical results show that adopting NRPM mitigates the influence of any datapoint on downstream predictions. Three paradigms of robust reprogramming are then introduced: one which only involves implementing the NRPM architecture, one which involves NRPM and fine-tuning hyperparameters, and one which involves NRPM and finetuning all parameters in the model. Experimental results show that these paradigms (and especially paradigm 3) can provide significant improvements over existing robustness techniques. For example, in ResNets trained on CIFAR10, paradigm 3 results in significantly higher robustness than state of the art adversarial training techniques, with only a modest drop in clean accuracy.

**Strengths:**

* There is a great deal of novelty in this approach. Many defenses against adversarial examples atttempt to learn weights that are more robust, or to detect/purify adversarial inputs to a network. I am not aware of any prior work that modifies the model of computation to increase robustness in this way.
* Code and model weights have been provided for replication purposes.
* The experimental results are impressive. In particular, in certain cases the third paradigm can increase both robustness and clean accuracy.

**Weaknesses:**

* Theorem 3.2 is presented in a confusing manner. You refer to $x_0$ as the perturbation, but isn't it instead the location of a possible perturbation?
* The theoretical justification for the robustness of NRPM largely relies on the influence functions that are derived for NRPM and LPM. However, there's no explicit connection that's made between influence functions and adversarial robustness. It's not immediately clear to me that the influence function on its own would immediately imply adversarial robustness. I think additional justification for this relationship would be necessary in the final version of this paper (or references to related works which establish this relationship).
* On a similar note, there are settings that have been studied in the literature in which an adversary is capable of making unbounded changes to the input (i.e. [1,2]). It seems that the influence function would be less informative about the robustness of the model to adversaries that are unbounded in $\ell_p$ space. I think it should be made more explicit if you are only envisioning this as a defense against $\ell_p$ bounded adversaries.
* The table and figure captions could use more information about the experimental setups. For example, its unclear which attacks/norms mentioned in section 5.1 are used in which table.

[1] Xiao, Chaowei, Jun-Yan Zhu, Bo Li, Warren He, Mingyan Liu, and Dawn Song. "Spatially transformed adversarial examples." arXiv preprint arXiv:1801.02612 (2018).
[2] Kaufmann, Max, Daniel Kang, Yi Sun, Steven Basart, Xuwang Yin, Mantas Mazeika, Akul Arora et al. "Testing robustness against unforeseen adversaries." arXiv preprint arXiv:1908.08016 (2019).

**Questions:**

* What is the difference in inference efficiency between a standard MLP and a reprogrammed MLP (in terms of memory usage/inference time)?
* Similarly, how does execution time scale with the depth/width of the network?
* Are there any adaptive attacks that could improve over AutoAttack against robust reprogramming? Does this approach just reduce to gradient masking?
* Is the NRPM approach suitable for training from scratch? If so, do you think that would that further improve over the performance of paradigm 3?
* Would an adversary with knowledge of the robust reprogramming scheme be able to design an adaptive attack to counteract these defenses?

---

> ### Author Response · Authors · 2024-11-22
>
> **W1:**  {${x_d}$}$_{d=1}^D$ represents the initial input, and $x_0$ denotes the perturbation introduced to the data. To avoid confusion, we have changed $x_0$ to $\Delta x$ and updated the notation accordingly in the revised version.
>
> **W2:**  The influence function measures the sensitivity of model predictions to introduced perturbations, which is highly correlated with adversarial robustness. Our theoretical justification builds on this perspective to further validate the robustness implications of our methods, in addition to empirical experiments.
>
> In fact, the influence function is well established in robust statistics and has recently been applied to adversarial robustness. Here, we list several papers [1][2][3] that highlight the close relationship between influence functions and adversarial robustness.
>
> References:
> 1. Lai, Lifeng, and Erhan Bayraktar. "On the adversarial robustness of robust estimators." IEEE Transactions on Information Theory 66.8 (2020): 5097-5109.
> 2. Deng, Zhun, et al. "Interpreting robust optimization via adversarial influence functions." International Conference on Machine Learning. PMLR, 2020.
> 3. Cohen, Gilad, Guillermo Sapiro, and Raja Giryes. "Detecting adversarial samples using influence functions and nearest neighbors." Proceedings of the IEEE/CVF conference on computer vision and pattern recognition. 2020.
>
> **W3:**  Thanks for your insightful comments. The influence function is introduced to measure the sensitivity of the model to the perturbation, which is useful no matter under bounded or unbounded $\ell_p$ adversaries. This work indeed focuses on $\ell_p$ bounded attack like PGD, FGSM, and AA, since they are the most commonly used evaluation.
>
> To validate the effectiveness of our method under unbounded attack, we include the experiments under DeepFool with different iterations, which do not restrict the bound on the attack budget. As shown in the following table, our method can outperform other baselines across all the iterations. When the number of iteration goes into extremely large (unbounded), all the model will have nearly zero accuracy.
>
> **Table 3: Adversarial robustness on CIFAR10 with ResNet18 as backbone.**
>
> | Model \ Iteration      | DeepFool-1 | DeepFool-2 | DeepFool-3 | DeepFool-5 |
> |------------------------|------------|------------|------------|------------|
> | PGD-AT                | 61.79      | 40.32      | 14.81      | 0.82       |
> | TRADES-2.0            | 66.35      | 55.99      | 25.91      | 3.45       |
> | TRADES-0.2            | 54.59      | 37.04      | 12.98      | 1.61       |
> | MART                  | 63.18      | 44.88      | 25.63      | 4.38       |
> | SAT                   | 61.45      | 53.67      | 22.34      | 2.12       |
> | AWP                   | 62.18      | 31.04      | 26.25      | 2.76       |
> | Consistency           | 63.54      | 47.63      | 21.27      | 1.63       |
> | DYNAT                 | 65.22      | 50.01      | 28.72      | 5.38       |
> | Paradigm 3 (Ours)     | 68.75      | 61.99      | 36.50      | 10.43      |
>
> Additionally, we believe the papers you mentioned [1][2] fall into the category of non-$\ell_p$ attacks, which we consider as a very interesting topic. We also conducted some experiments in this area earlier, but we did not include them in the initial submission. As shown in the table below, our NRPM architecture demonstrates improvements over the vanilla LPM in most cases.
>
> **Table 4: Robustness under non-$\ell_p$ attack on CIFAR-10**
>
> | Type      | Model | Level 1 | Level 2 | Level 3 |
> |-----------|-------|---------|---------|---------|
> | Blur      | LPM   | 32.43   | 34.62   | 37.50   |
> |           | NRPM  | 38.46   | 40.50   | 43.13   |
> | Snow      | LPM   | 77.62   | 58.34   | 54.51   |
> |           | NRPM  | 76.80   | 61.58   | 57.74   |
> | Frost     | LPM   | 78.61   | 69.45   | 54.86   |
> |           | NRPM  | 77.82   | 68.58   | 56.52   |
> | Impulse   | LPM   | 75.00   | 64.11   | 52.96   |
> |           | NRPM  | 75.72   | 64.40   | 54.28   |
> | Shot      | LPM   | 72.61   | 61.71   | 51.99   |
> |           | NRPM  | 75.65   | 66.80   | 48.65   |
>
>
>
> **W4:**  Thanks for pointing out, we have included more information in the revised version.

---

> ### Author Response · Authors · 2024-11-22
>
> **Q1:** Following your suggestion, we also include a comparison of the inference running time between the vanilla LPM and our NRPM in the table below. As shown, our NRPM incurs only 2-4 times the inference cost of the LPM, which we consider acceptable.
>
> **Table 5: Running time with MLP as backbone.**
>
> | Depth (Layers) | 2      | 3      | 4      | 5      | 6      |
> |----------------|--------|--------|--------|--------|--------|
> | LPM (ms)       | 178.61 | 198.47 | 212.35 | 225.72 | 232.65 |
> | NRPM (ms)      | 404.83 | 587.19 | 755.62 | 944.46 | 1096.93 |
>
> Additionally, we want to point out that our main contribution lies in proposing a novel approach to improve robustness, which is orthogonal to existing methods. Regarding efficiency, while we acknowledge that our proposed architecture introduces additional, yet acceptable, computational costs, our robustness reprogramming improves robustness in a lightweight manner, either without tuning or with slight tuning. Most importantly, we want to emphasize again the significant contributions of our work to the current field of robust machine learning, as highlighted at the beginning of our response.
>
>
>
> **Q2:** Refer to Q1.
>
> **Q3:** We believe our evaluation is already adaptive and comprehensive enough to avoid the issue of gradient masking:
>
> - In our proposed NRPM layer, there is no non-differentiable operator, which mitigates concerns about differentiability issues.
> - Our evaluation using AutoAttack already includes the black-box square attack, which does not rely on gradients.
> - In Section 5.4, we also evaluate the certified robustness using randomized smoothing, which theoretically guarantees robustness under any attack.
> - We include a visualization analysis of the hidden embeddings in Section 5.3 to further validate the robustness of our proposed robust layer.
>
>
>
> **Q4:** The NRPM approach is suitable for training from scratch, but it cannot improve over Paradigm 3. As shown in the table, NRPM sacrifices clean performance, which limits its ability to enhance robustness. In contrast, our Paradigm 3 can simultaneously learn the optimal combination of {${\lambda}$} and model parameters to improve both natural and robust performance, outperforming NRPM. Certainly, we believe that training a hybrid model from scratch would be comparable to Paradigm 3, but the computational cost would be prohibitively expensive. This further highlights the necessity of our proposed robustness reprogramming.
>
> **Table 6: Adversarial robustness on MNIST with 3-layer MLP as backbone.**
>
> | Method / Budget | Natural | 0.05  | 0.1   | 0.15  | 0.2   | 0.25  | 0.3   |
> |------------------|---------|-------|-------|-------|-------|-------|-------|
> | NRPM            | 78.7    | 67.0  | 64.3  | 59.0  | 49.9  | 43.7  | 35.2  |
> | Paradigm 3      | 86.1    | 81.7 | 75.8 | 66.7 | 58.7 | 50.1 | 39.8 |
>
>
> **Q5:** Refer Q3.

---

> > ### Comment · Reviewer_ktRd · 2024-11-26
> >
> > Thank you for your thorough response. I appreciate the additional results and clarifications. I will be raising my score accordingly.

---

> > > ### Author Response · Authors · 2024-11-26
> > >
> > > Thank you for your feedback! We truly appreciate your recognition of our work.

---

### Official Review · Reviewer_Pa5t · 2024-11-05

**Soundness:** 3
**Presentation:** 2
**Contribution:** 3
**Rating:** 6
**Confidence:** 3

**Summary:**

This paper proposes a "reprogramming" approach to improve the robustness of neural networks with minimal or no alteration of the model parameters. The reprogramming approach is executed by incorporating additional set of parameters (one per input dimension) in each layer to reweight the importance of features at inference time. The approach is motivated from a perspective of looking at the function of each unit as a linear feature pattern matching that is proposed to be replaced by a nonlinear version. The final implementation of the approach combines the activity of both linear and nonlinear feature pattern matching methods.

**Strengths:**

1. Proposed idea of reprogramming is interesting as it would allow reusing the learned features by the original model. May be an important approach to explore in large models.
2. Empirical results consider multiple datasets of various sizes and different perturbations

**Weaknesses:**

1. The approach is based on eq 1 and the precursor to this for what is named as "linear feature pattern matching". The first equation (unnumbered equation before eq 1) formulates the OLS solution as the minimum of \mathcal{L}=\sum_{d=1}^{D}{(\frac{y}{D}-a_d.x_d)^2}. However, this formulation of the solution assumes that the prediction error is distributed uniformly across all dimensions which is not generally true and may indeed be very constraining in certain situations. Despite this possible issue, I didn't see any mention of this anywhere in the paper which in my opinion is a big weakness.

2. Some of the key implementation details of the approach were unclear. What data is used to optimize the $w_i$ parameters? Are they optimized over the full training set of each dataset? Alg 1 contains a loop of K iterations for each $x_d$. Does this mean that the approach finds independent $w_d$ for each sample independently? In what order are different parameters/hyperparameters optimized? A step by step description of the approach or a pseudo-code would be helpful if not necessary.

3. The approach requires computing additional activations for every layer that may be costly but no comparison of runtime, computation cost etc is provided.

4. Details of Adv-train baseline is missing. Was this the result of adversarial training from scratch or fine-tuning?

5. In Table5, the AA accuracy is substantially higher than that of PGD (64.60 vs. 57.23). This to me looks like a big red flag. AA is an ensemble attack made of 4 separate attacks including two effective variations of PGD. Because of this, AA accuracy is always lower than that of PGD. I'm doubtful of the validity of empirical evaluations.

**Questions:**

1. Related to weakness #1 above, on line 167 it is stated that by replacing the loss function with the proposed one the impact of outliers will be reduced, referring to Huber and Ronchetti's book Robust Statistics. The connection is unclear to me and I think deserves a more detailed explanation. Can you expand?

2. Line 105, by "robust training" do you mean "adversarial training"?

3. Line 244: "One naive approach is to simply replace the vanilla linear pattern matching (LPM) with NRPM. However, this naive approach does not work well in practice". Why doesn't this approach work?

4. Line 231: according to the text, $x_o$ is not the perturbation but the initial input. Is that correct?

5. In table1 and others, what are the numbers at the top of each column? Are they the epsilon values used for test? What attack was used?

6. Adaptation of the approach to CNNs needs more explanation. Were separate $w_i$ parameters considered per filter or per activation?

7. Fig 5: likelihood in the bar plots doesn't look like proper probability density functions. What is being shown?

8. Fig 5 caption: 1 and 2 correspond to (a) and (b) in the figure?

---

> ### Author Response · Authors · 2024-11-22
>
> **W1:** In the first equation, no assumption is made about the prediction error. The OLS formulation $L(z)$ is directly derived from the inverse optimization problem of the linear feature transformation $z = \mathbf{a}^\top \mathbf{x} = \sum_{d=1}^D a_d \cdot x_d$. Specifically, the first-order optimality condition, $\frac{\partial L(z)}{\partial z} = 0,$ yields the linear transformation $z^* = \sum_{d=1}^D a_d \cdot x_d.$
>
> **W2:**
> - The $w_d$ is not the parameter to be learned; instead, it is computed as $w_d^{(k)} = \frac{a_d x_d^{(k)}}{\sum_{d=1}^D a_d x_d^{(k)}}$ at the $k$-th layer in the neural network, where $x_d$ represents the input value, $a_d$ is the parameter to be learned, and $z^{(k)}$ is the hidden embedding at the $k$-th layer.
>
> - In Alg 1, the entire {${x_d}$}$_{d=1}^D$ is one data sample, and $d$ is the index of dimension. So the approach will find independent $w_d$ for each dimension $d$ in each sample.
>
> - The parameters/hyperparameters in the model to be optimized include two parts: (1) model parameters {${a_d}$}$_{d=1}^D$, same as backbone model, (2) balance hyperparameter {${\lambda}$} introduced in hybrid parameters. The parameters can be optimized into two phases:
>   - Phase 1 (Pre-training): the model parameters {${a_d}$}$_{d=1}^D$ is pre-trained and saved in backbone model.
>   - Phase 2 (Fine-tuning): this phase can be divided into three paradigms as in Section 4, including manually selecting the hyperparameter {${\lambda}$} with fine-tuning (paradigm 1); only fine-tuning {$\lambda$} while freezing the {${a_d}$}  (paradigm 2); fine-tuning both the {${a_d}$} and {${\lambda}$} at the same time (paradigm 3).
>
> - The pseudo-code for implementation details have been included in the Alg 2 in Appendix A.1 due to space limit in the submission. Moreover, we have provided the available code in [https://anonymous.4open.science/r/NRPM-322C/](https://anonymous.4open.science/r/NRPM-322C/).
>
> **W3:** Following your suggestion, we also include a comparison of the inference running time between the vanilla LPM and our NRPM in the table below. As shown, our NRPM incurs only 3-4 times the inference cost of the LPM, which we consider acceptable.
>
> **Table 1: Running time with ResNet as backbone.**
>
> | Layers  | 10     | 18     | 26     | 34     | 42     | 50     | 58     | 66     | 74     |
> |---------|--------|--------|--------|--------|--------|--------|--------|--------|--------|
> | LPM (ms)| 27.81  | 47.10  | 66.50  | 87.18  | 108.78 | 125.65 | 130.08 | 142.42 | 159.56 |
> | NRPM (ms)| 88.53 | 167.21 | 253.63 | 338.78 | 416.78 | 497.46 | 552.18 | 628.35 | 732.67 |
>
> Additionally, we want to point out that our main contribution lies in proposing a novel approach to improve robustness, which is orthogonal to existing methods. Regarding efficiency, while we acknowledge that our proposed architecture introduces additional, yet acceptable, computational costs, our robustness reprogramming improves robustness in a lightweight manner, either without tuning or with slight tuning. Most importantly, we want to emphasize again the significant contributions of our work to the current field of robust machine learning, as highlighted in our common response.
>
>
> **W4:**  We compare the baselines including: PGD-AT [1], TRADES [2], MART [3] and SAT [4] and AWP [5]. For fair comparison, we conduct adversarial training on the baseline from scratch. We apply the same training setting as ours by setting training batch size as 128, weight decay as 2e-5, momentum as 0.9. We train the models with 200 epochs with adjusted learning rate (0.1 for 0-100 epochs; 0.01 for 100-150 epochs; 0.001 for 150-200 epochs). For TRADES, TRADES-2.0 and TRADES-0.2 means that we set the sensitivity of regularization hyperparameter $\beta$ in TRADES as 2.0 and 0.2, respectively.
>
> References:
> 1. A. Madry, A. Makelov, L. Schmidt, D. Tsipras, and A. Vladu, "Towards deep learning models resistant to adversarial attacks.", 2017.
> 2. H. Zhang, Y. Yu, J. Jiao, E. Xing, L. El Ghaoui, and M. Jordan, "Theoretically principled trade-off between robustness and accuracy.", 2019.
> 3. Y. Wang, D. Zou, J. Yi, J. Bailey, X. Ma, and Q. Gu, "Improving adversarial robustness requires revisiting misclassified examples.", 2020.
> 4. L. Huang, C. Zhang, and H. Zhang, "Self-adaptive training: beyond empirical risk minimization.", 2020.
> 5. D. Wu, S.-T. Xia, and Y. Wang, "Adversarial weight perturbation helps robust generalization.", 2020.
>
>
> **W5:**  Thank you for pointing that out. We apologize for the mistake made while copying the results of AutoAttack. We have double-checked all the experimental results and have updated the correct values in the revised version. The conclusions remain the same after correcting the results.

---

> ### Author Response · Authors · 2024-11-22
>
> **Q1:**  This technique is a well-established theory in robust statistics, so we will briefly explain it here. I am happy to provide more details as follows:
>
> - Intuitively, when adversaries perturb the input, the residual $z / D - a_d \cdot x_d$ tends to increase. Consequently, the quadratic penalty on the residual will dominate the Ordinary Least Squares (OLS) estimator, resulting in a shift towards those dominating input pixels and significantly impacting the representation of the output. By replacing the quadratic penalty (OLS estimator) with a linear alternative (LAD estimator) on the residual $z / D - a_d \cdot x_d$, the impact of outliers can be mitigated. The robustness of OLS and LAD estimators has been well validated in the field of robust statistics.
>
> - For a simpler example, consider the mean estimator as the OLS estimator and the median estimator as the LAD estimator. If there are perturbations in the data points, the median is generally more robust and less sensitive to such perturbations.
>
>
> **Q2:** Yes.  ”robust training”  means ”adversarial training”.
>
> **Q3:** Simply replacing LPM with NRPM will result in clean performance drop. As shown in the following table, when $\lambda$ is set to 0, our model becomes equivalent to NRPM, leading to a reduction in natural clean performance to 18.8%. This degradation in clean performance can be significantly mitigated through fine-tuning, which improves the natural performance to 78.7%. However, this is still not the optimal case. As demonstrated by Paradigm 3 (tuning all), we can simultaneously learn the optimal combination of {${\lambda}$} and model parameters to enhance both natural and robust performance. Therefore, it is necessary to construct a hybrid architecture and learn the optimal combination of LPM and NRPM.
>
> **Table: Robustness reprogramming under 3 paradigms on MNIST with 3-layer MLP as backbone.**
>
> | Method / Budget          | Natural | 0.05  | 0.1   | 0.15  | 0.2   | 0.25  | 0.3   | [$\lambda_1, \lambda_2, \lambda_3$] |
> |--------------------------|---------|-------|-------|-------|-------|-------|-------|-----------------------------------|
> |                          | 90.8    | 31.8  | 2.6   | 0.0   | 0.0   | 0.0   | 0.0   | [1.0, 1.0, 1.0]                  |
> |                          | 90.8    | 56.6  | 17.9  | 8.5   | 4.6   | 3.0   | 2.3   | [0.9, 0.9, 0.9]                  |
> |                          | 90.4    | 67.1  | 30.8  | 17.4  | 10.6  | 6.5   | 4.5   | [0.8, 0.8, 0.8]                  |
> |                          | 89.7    | 73.7  | 43.5  | 25.5  | 16.9  | 11.7  | 9.2   | [0.7, 0.7, 0.7]                  |
> | Paradigm 1  | 88.1 | 75.3  | 49.0  | 31.0  | 22.0  | 15.5  | 12.4  | [0.6, 0.6, 0.6]                  |
> |  (without tuning)     | 84.1    | 74.4  | 50.0  | 31.9  | 22.8  | 18.1  | 14.3  | [0.5, 0.5, 0.5]                  |
> |                          | 78.8    | 70.4  | 48.3  | 33.9  | 24.1  | 18.4  | 14.6  | [0.4, 0.4, 0.4]                  |
> |                          | 69.5    | 62.6  | 45.2  | 31.5  | 23.1  | 19.0  | 15.5  | [0.3, 0.3, 0.3]                  |
> |                          | 58.5    | 53.2  | 38.2  | 27.6  | 22.2  | 16.4  | 12.9  | [0.2, 0.2, 0.2]                  |
> |                          | 40.7    | 38.3  | 29.7  | 22.8  | 16.8  | 12.9  | 11.1  | [0.1, 0.1, 0.1]                  |
> | NRPM (without tuning)    | 18.8    | 17.6  | 16.4  | 14.6  | 12.4  | 10.7  | 9.4   | [0.0, 0.0, 0.0]                  |
> | NRPM (tuning all)        | 78.7    | 67.0  | 64.3  | 59.0  | 49.9  | 43.7  | 35.2  | [0.0, 0.0, 0.0]                  |
> | Paradigm 2 (tuning $\lambda$) | 81.5 | 75.3  | 61.2  | 44.7  | 33.7  | 26.0  | 20.1  | [0.459, 0.033, 0.131]           |
> | Paradigm 3 (tuning all)  | 86.1    | **81.7** | **75.8** | **66.7** | **58.7** | **50.1** | **39.8** | [0.925, 0.119, 0.325]           |
>
>
> **Q4:** The {${x_d}$}$_{d=1}^D$ is the initial input, and $x_0$ represents the perturbation introduced to the data. To avoid confusion, we have changed $x_0$ to $\Delta x$ and updated the notation accordingly in the revised version.
>
> **Q5:** The numbers at the top of each column are the budgets of attack. The attacks used are FGSM (Table 1 & 2) and PGD (Table 4). The budgets are both measured with the $L_\infty$ norm.
>
> **Q6:** $w_i$ is not a parameter to be learned but is computed from features $x_d$ and pre-trained model parameters $a_d$. The only parameter to be learned is $\lambda$. Each layer has a single parameter $\lambda$ that needs to be learned.
>
> **Q7:** The bar plots show the probabilities of the data being classified into different categories by the models.
>
>
> **Q8:** Thank you for pointing that out. (1) and (2) correspond to (a) and (b). We have corrected the errors in the updated version.

---

> > ### Comment · Reviewer_Pa5t · 2024-11-26
> > **thanks**
> >
> > I appreciate the responses. Most of my questions are answered but see below:
> >
> > Re W1: expanding on what I wrote in my initial review. I believe the equation on page 3 got me confused. The cost function associated with OLS is of the form $\mathcal{L}=\lVert y - \mathbf{x}\mathbf{\theta} \rVert = \Big( y - \sum_{d=1}^{D}{x_d.\theta_d\Big)^2}$. But the form considered in that equation is $\mathcal{L}=\sum_{d=1}^{D}{(\frac{z}{D}-a_d.x_d)^2}$ which I took as a solution which equal parts of the response variable $y$ above ($z$ in the paper) is to be explained by each predictor dimension $x_d$. The authors' response unfortunately didn't help in clarifying that point.

---

> > > ### Author Response · Authors · 2024-11-27
> > >
> > > Thank you for your insightful feedback, which helps us better understand your question. In fact, our formulation is different from a traditional OLS problem, and we will change this name in our revision to avoid confusions. We would like to clarify as follows:
> > >
> > > A conventional OLS problem,
> > > $L(\theta) = \left( y - \sum_{d=1}^{D} x_d \cdot \theta_d \right)^2,$
> > > aims to optimize the parameter $\theta$. However, optimizing $y$ in this formulation does not make any sense because no matter how we change the penalty (to $L_1$ or other robust alternatives),
> > > $y = \sum_{d=1}^{D} x_d \cdot \theta_d$
> > > is always the optimal solution. Therefore, conventional OLS cannot help solve our problem.
> > >
> > > What we proposed in this work is a totally new formulation (we do not find this perspective in the literature):
> > > $\mathcal{L}(z) = \sum_{d=1}^{D} \left( \hat{z}_d - a_d \cdot x_d \right)^2,$
> > > where we optimize the embedding $z$. The key of this new formulation is to separate the impact of each feature dimension, and the impact of each dimension $x_d$ is relevant to the coefficient $a_d$ and the chosen penalty, as illustrated in our final solution in Equation (3) in page 4.
> > >
> > > To summarize, our method is fundamentally different from the traditional OLS problem: the two formulations are different, and they aim to optimize different variables. We appreciate your valuable comments. We will correct the name and avoid confusions in the revised version. We hope this can resolve your concern. Please kindly let us know if any further clarification is needed.

---

> > > > ### Comment · Reviewer_Pa5t · 2024-11-28
> > > >
> > > > Thanks for the additional clarification. I encourage the authors to update their submission to reflect the changes while there is still time. I increased my score by 1 point to acknowledge the improvements.

---

> > > > > ### Author Response · Authors · 2024-12-03
> > > > >
> > > > > Thank you for your valuable feedback! We have updated the name in the revised version to avoid any confusion. Additionally, we would greatly appreciate your insights on any remaining weaknesses so we can further refine our efforts and improve the paper based on your suggestions!

---

### Author Response · Authors · 2024-11-22
**[Common Response] Statement of our significant contributions in robust machine learning**

We sincerely thank the reviewers for their constructive feedback and valuable suggestions. We have carefully addressed all concerns and provided a revised manuscript. Below, we provide an overall response to highlight the **significant contributions, reliable evaluations, and computation cost** of our work.

While adversary training-based methods have emerged as the mainstream approaches for  improving robustness, they face a clear performance ceiling, substantial training overhead, and the risk of overfitting.
Our proposed work provides a  transformative breakthrough that is orthogonal to, and compatible with, existing adversarial training techniques.
Unlike traditional methods, our approach is a flexible reprogramming framework that can be applied to any pre-trained model—adversarially or normally trained—without requiring complete retraining to further boost the robustness with reduced training costs.

It is widely recognized that many adversarial defenses strategies suffer from a false sense of security [1].
To ensure the robustness in our work are reliable and not subject to a false sense of security, we conducted comprehensive evaluations, including multiple adaptive white-box adversarial attacks and certified robustness assessments. These evaluations demonstrate significant and consistent improvements over existing adversarial training baselines. In additional to comprehensive empirical evaluations, we also demonstrate how the proposed work can improve robustness with rigorous theoretical convergence and robustness analyses.


Regarding efficiency, we acknowledge that the proposed Non-linear Robust Pattern Matching (NRPM) incurs a constant-factor increase in inference computation costs compared to the vanilla models. However, this trade-off is acceptable and justified by the substantial robustness improvements achieved. Additionally, our robustness reprogramming framework can significantly reduce training costs, enabling improved robustness with pre-trained models without the need for extensive tuning or requiring only slight adjustments.
While efficiency is not the primary focus of this work, we recognize its importance and plan to accelerate NRPM in future research.

To summarize, this work provides a revolutionary perspective on improving adversarial robustness through reprogramming, validated by both theoretical analyses and comprehensive empirical results. We are confident that this approach not only complements adversarial training but also opens new doors for developing robust deep learning models.

We hope these clarifications and revisions address the reviewers' concerns and demonstrate the value of our contribution to the community. Please kindly let us know if there are any further remaining concerns, and we are happy to clarify.

[1] Carlini, N., and D. Wagner. "Obfuscated gradients give a false sense of security." ICML, 2018.

---

### Author Response · Authors · 2024-11-25
**Request for Feedback on Our Rebuttal**

Dear reviewers,

We hope this message finds you well. Since the rebuttal period is approaching its end, we want to kindly remind you that we have submitted our rebuttal addressing your comments and concerns regarding our paper.

We greatly value your feedback and have carefully considered each point in our response. If you have had a chance to review the rebuttal, we would appreciate knowing whether it adequately resolves your concerns.

Please feel free to let us know if there are any remaining issues or further clarifications needed. We are happy to provide additional information to ensure the manuscript meets the required standards.

Thank you once again for your time and effort in reviewing our work.

Best regards,
All authors

---

### Meta-Review · Area_Chair_f8Ka · 2024-12-20

**Metareview:**

This paper presents Robustness Reprogramming, a novel method for enhancing the adversarial robustness of pre-trained models without modifying their core parameters. By introducing Nonlinear Robust Pattern Matching (NRPM) and three reprogramming paradigms, the authors propose an approach that is orthogonal to traditional adversarial training. Their method combines theoretical rigor with comprehensive experimental validation, demonstrating consistent robustness improvements across various models and datasets.

The reviewers unanimously recommend acceptance, recognizing the paper's novelty and solid empirical results. While some initial concerns were raised about the theoretical framing, notation clarity, and scalability, the authors effectively addressed these through detailed responses, revisions, and additional experiments on CIFAR-100, Tiny-ImageNet, and WideResNet28-10. These efforts strengthened the submission significantly for more real-world settings.

Overall, this paper is a strong submission, offering a new direction for robust machine learning and practical utility for adapting pre-trained models. I recommend acceptance as a spotlight, given its innovative approach and potential impact.

**Additional Comments On Reviewer Discussion:**

During the review discussion, the reviewers focused on clarity in theoretical justifications, the trade-off in clean performance observed in Paradigm 1, and scaling the method to larger datasets. The authors clarified the theoretical framework, improved the notation, and demonstrated that Paradigms 2 and 3 effectively balance clean and robust performance. Additional experiments with larger datasets addressed scalability concerns.

The reviewers uniformly appreciated the depth of the authors’ responses and the improvements made to the manuscript. Their unanimous recommendation reflects the paper's high quality and its contribution to advancing robustness research. This submission stands out as a strong and well-rounded piece of work.

---

### Decision · Program_Chairs · 2025-01-22

Accept (Spotlight)